

# Spatial and temporal variations in rockwall erosion rates around Pigne d'Arolla, Switzerland, derived from cosmogenic [10]Be in medial moraines at five adjacent valley glaciers

Katharina Wetterauer[1] and Dirk Scherler[1,2]

[1]Earth Surface Geochemistry, GFZ German Research Centre for Geosciences, 14473 Potsdam, Germany
[2]Institute of Geographical Sciences, Freie Universität Berlin, 12249 Berlin, Germany

*Correspondence to*: Katharina Wetterauer (katharina.wetterauer@gfz-potsdam.de)

**Abstract.** Rockwall erosion in high-alpine glacial environments varies both temporally and spatially. Where rockwalls flank glaciers, changes in debris supply and supraglacial cover will modify ice ablation. Yet, quantifying spatiotemporal patterns in

erosion across deglaciating rockwalls is not trivial. At five adjacent valley glaciers around Pigne d'Arolla in Switzerland, we derived apparent rockwall erosion rates using [10]Be cosmogenic nuclide concentrations ([[10]Be]) in medial moraine debris. Systematic downglacier-sampling of six medial moraines that receive debris from rockwalls with differing orientation, slope and deglaciation histories enabled us to assess rockwall erosion through time and to investigate how distinct spatial source rockwall morphology may express itself in medial moraine [[10]Be] records. Our dataset combines 24 new samples from medial

moraines of Glacier du Brenay, Glacier de Cheilon, Glacier de Pièce, and Glacier de Tsijiore Nouve, with 15 published samples from Glacier d'Otemma. For each sample, we simulated the glacial debris transport using a simple debris particle trajectory model, to approximate the time of debris erosion and to correct the measured [[10]Be] for post-depositional [10]Be accumulation. Our derived apparent rockwall erosion rates range between ~0.6 and 10.0 mm yr[-1]. Whereas the longest downglacier [[10]Be] record presumably reaches back to the end of the Little Ice Age (LIA) and suggests a systematic increase in rockwall erosion

rates over the last ~200 years, the shorter records only cover the last ~100 years from the recent deglaciation period and indicate temporally more stable erosion rates. For the estimated time of debris erosion, ice cover changes across most source rockwalls were small, suggesting that our records are largely unaffected by the contribution of recently deglaciated bedrock of possibly different [[10]Be]; but admixture of subglacially derived debris cannot be excluded at every site. Comparing our sites suggests that apparent rockwall erosion rates are higher where rockwalls are steep and north-facing, indicating a potential slope and

temperature control on rockwall erosion around Pigne d'Arolla.



## 1 Introduction

Alpine glacial environments are characterized by steep rockwalls near the head of valley glaciers that expose bare rock surfaces to erosion. Erosion in these environments largely proceeds via frequent small-scale rock falls and infrequent large-scale mass wasting processes (e.g., Boulton and Deynoux, 1981; Anderson, 2000; Arsenault and Meigs, 2005). On glacier surfaces,

rockwall debris is passively entrained, becoming part of the glacial system, e.g., in the form of medial moraines (e.g., Eyles and Rogerson, 1978; Gomez and Small, 1985; Anderson, 2000). Where debris is transported on the surface, it alters sub-debris melt-rates (Østrem, 1959) and potentially modifies glacier retreat (e.g., Scherler et al., 2011; Rowan et al., 2015; Vincent et al., 2016). The debris cover thickness and its change are influenced by the rate at which the surrounding rockwalls erode (Scherler and Egholm, 2020). However, rockwall erosion, and thus, debris supply rates are not spatially uniform, as evidenced

by topographic asymmetry across glacial landscapes (e.g., Gilbert, 1904; Tuck, 1935; Naylor and Gabet, 2007). Although such asymmetry is commonly associated with spatial and temporal gradients in erosion, the underlying conditions are still controversial.

In the European Alps, recent temporal and spatial variability in rockwall erosion has been commonly linked to post-Little Ice

Age (LIA; post ~1850) warming and/or to locally distinct temperature-related conditions. As temperatures increase and permafrost thaws, the stability of alpine rockwalls decreases, promoting rock falls and slope failures (e.g., Gruber and Haeberli, 2007; Huggel et al., 2010; Ravanel et al., 2010; Fischer et al., 2012). Enhanced destabilization is also observed in recently deglaciating bedrock, where glacial thinning and unloading are thought to affect the thermo-mechanical stress field and frost-damage intensity within the adjacent bedrock (e.g., Wegmann et al., 1998; Kenner et al., 2011; Hartmeyer et al., 2020). Spatial

variability in erosion has previously been related to rock face aspect, and a higher rock fall activity at north-facing rockwalls has been associated with differences in moisture supply and damage by frost (e.g., Coutard and Francou, 1989; Gruber et al., 2004; Sass, 2005, 2010). Yet, rock falls are pronounced stochastic processes (e.g., Ward and Anderson, 2011; Sanders et al., 2013), and their observation is typically based on repeated monitoring over comparatively short time periods ($10^0$-$10^1$ years). Therefore, the number of quantified rockwall erosion rates and their spatiotemporal analysis is still limited.


An alternative approach for quantifying rockwall erosion rates over longer time periods (>$10^2$-$10^4$ years) uses downglacier records of *in situ*-produced cosmogenic $^{10}$Be concentrations ([$^{10}$Be], atoms g$^{-1}$) in medial moraines. By interval sampling along medial moraines, rockwall erosion is quantified through time, based on the [$^{10}$Be] within the debris that reflects the rockwall erosion products. The approach exploits the conveyor-belt nature of glaciers: passively transported rockwall debris forms

medial moraines as it melts out below the equilibrium line altitude (ELA) in the glacier ablation zone or as it is merged from individual glacier branches (e.g., Eyles and Rogerson, 1978; Gomez and Small, 1985; Anderson, 2000). Thus, medial moraine deposits tend to be older downglacier (for detailed treatments on rockwall [$^{10}$Be] in medial moraine debris see Ward and Anderson, 2011; Scherler and Egholm, 2020; Wetterauer et al., 2022a). Previous work from the Himalaya (Scherler and



Egholm, 2020) and the Swiss Alps (Wetterauer et al., 2022a) suggests that medial moraines can indeed act as archives for the temporal evolution of rockwall erosion. At both sites, downglacier records indicate an acceleration in erosion rates with climate warming, consistent with the aforementioned monitoring observations. However, using [¹⁰Be] from supraglacial debris has also proven challenging: medial moraine [¹⁰Be] likely reflect a mixed signal of rockwall erosion, rockwall deglaciation, and englacial/supraglacial transport time, and our understanding of how these processes affect erosion rate estimates is still limited.

This study examines patterns of rockwall erosion in a broader context using medial moraine [¹⁰Be] records from five adjacent valley glaciers surrounding the Pigne d'Arolla, a Swiss mountain peak in the Canton of Valais. We aim to (i) further evaluate the systematics of medial moraine [¹⁰Be]-derived estimates of rockwall erosion rates, (ii) check for temporal trends along downglacier medial moraine records, and (iii) test for differences among rockwall erosion rates in relation to spatially distinct geomorphic rockwalls. This study expands the work by Wetterauer et al. (2022a) on the Glacier d'Otemma, closely resembling their approach to remain as comparable as possible. Our new rockwall erosion dataset comprises downglacier [¹⁰Be] records from four new study sites: Glacier du Brenay, Glacier de Cheilon, Glacier de Pièce, and Glacier de Tsijiore Nouve, all flowing down either from the northern or southern flanks of the Pigne d'Arolla and receiving debris from rockwalls of varied exposure and morphology. We corrected our [¹⁰Be] records for englacial/supraglacial transport and derived estimates of apparent rockwall erosion rates, using a simple debris particle trajectory model (Wetterauer et al., 2022a,b), for which we generated ~40 year records of glacier surface velocities by manually tracing medial moraine boulders across orthoimages. Finally, we compared the different source rockwalls with respect to their area, elevation, slope, aspect and deglaciation history, based on former glacier outlines and historical photographs.

## 2 Study area

### 2.1 Pigne d'Arolla massif

The Pigne d'Arolla (3790 m; all elevations stated as m above sea level) is a mountain in the southern Swiss Alps, near the border to Italy (Fig. 1a,b). The massif surrounding it is part of the Dent Blanch nappe (Austroalpine unit) and consists of crystalline rocks of the Arolla series, mainly orthogneiss and metadiorites (swisstopo, 2022). The majority of its ice-free slopes are around 3000 m elevation, with more south- than north-facing slopes in terms of area (Fig. 2a,b). At this elevation and higher, slopes are typically inclined by 30 to 50°, with north faces tending to be steeper than south faces (Fig. 2a,c). At present, the area around Pigne d'Arolla is still glaciated (Fig. 1). Several valley glaciers emerge from the faces of an east-west trending ridgeline, which connects Pigne d'Arolla to the adjacent Mont Blanc de Cheilon, and flow downvalley in a northerly or southerly direction. These include the five study sites: Glacier du Brenay, Glacier de Cheilon, Glacier d'Otemma, Glacier de Pièce, and Glacier de Tsijiore Nouve. From now on, we refer to this set of five adjacent glacier catchments as the "Pigne d'Arolla massif" and to the individual glaciers as "Brenay", "Cheilon", "Otemma", "Pièce", and "Tsijiore Nouve", for brevity.



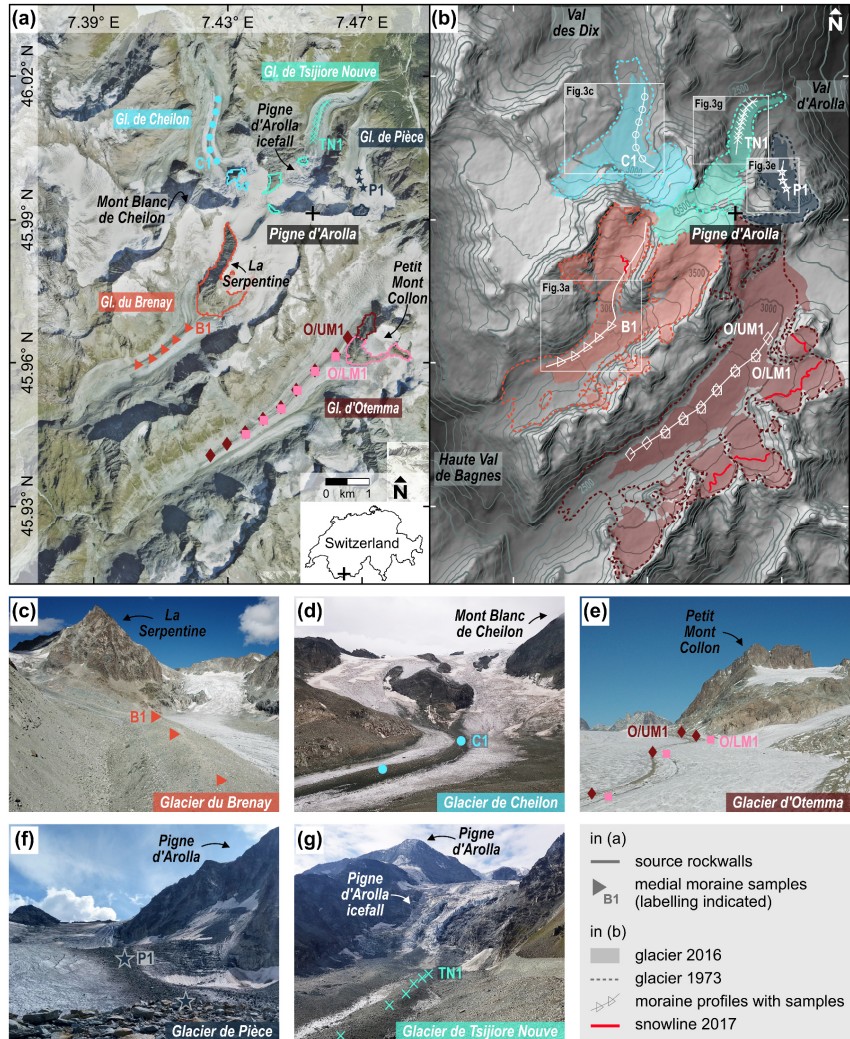

Figure 1: Pigne d'Arolla massif, Switzerland. (a) Orthoimage of the year 2017 showing the five adjacent glacier catchments, the respective medial moraine sample locations and their associated source rockwalls (orthoimage by swisstopo, 2022). (b) Hillshade image of the same area as in (a) with elevation as greyscale (bright = high). Coloured polygons/outlines show the glacier extents in 2016 and 1973 after Linsbauer et al. (2021) and Müller et al. (1976), respectively. The snowline of the 2017 orthoimage is indicated as approximation of the recent equilibrium line altitude. Contour lines are spaced by 100 m and based on the recent DEM. White rectangles indicate map extents shown in Fig. 3. (c-g) Field photographs showing medial moraines, approximated sample locations and associated source rockwalls of (c) Glacier du Brenay, (d) Glacier de Cheilon, (e) Glacier d'Otemma (samples from Wetterauer et al., 2022a), (f) Glacier de Pièce, and (g) Glacier de Tsijiore Nouve. Note the east-west trending ridgeline between Pigne d'Arolla and Mont Blanc de Cheilon in (a) following closely the ~3500 m contour line in (b).



Throughout the Pigne d'Arolla massif, rockwalls of varying extent and morphology deposit debris on the glacier surfaces, forming distinct medial moraines as the debris is transported downglacier (Fig. 1c-g). At present, these rockwalls are still located within the distribution zone of modelled mountain permafrost in the Swiss Alps (BAFU, 2005). The recent ELA is located above 3000 m (e.g., >3100 m at Otemma between autumn 2019 and 2021; GLAMOS, 2021b). Therefore, where recent debris deposition occurs below 3000 m elevation, in the ablation zone, downglacier debris transport is exclusively supraglacial.

However, aerial images form 1983 indicate a lower ELA and debris deposition in the accumulation zone above and, therefore, englacial transport in the past (swisstopo, 2022).

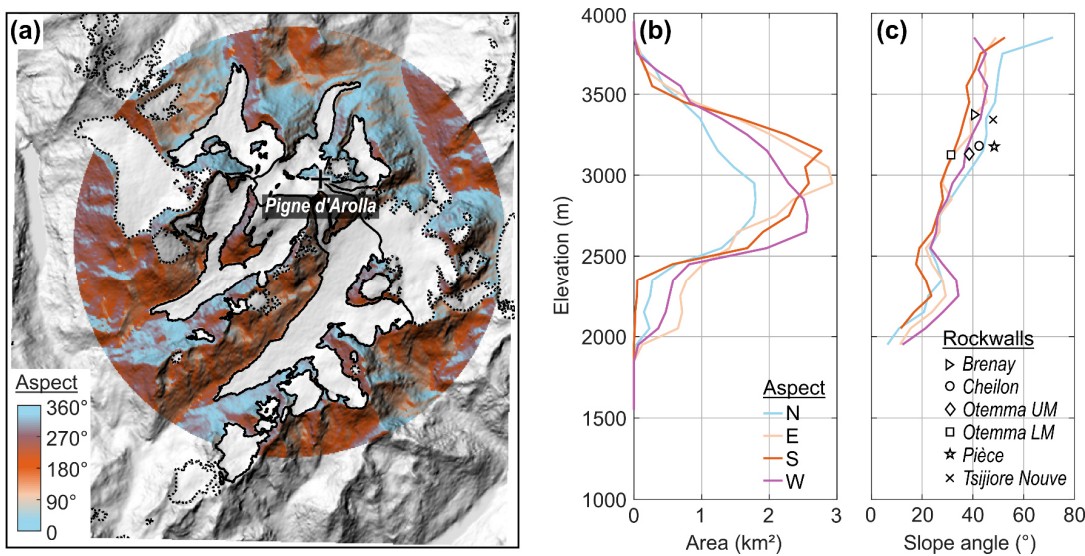

**Figure 2: Morphology of ice-free rock faces and slopes in an area with 6 km radius, centred on the studied glaciers of the Pigne d'Arolla massif. (a) Topographic map indicating the aspect of ice-free areas. Areas covered by glacial ice (dashed black outlines) are**
**excluded from the analysis and the five studied glacier sites (bold black outline) explicitly highlighted (2016 glacier outlines by Linsbauer et al., 2021). (b) Hypsometry and (c) slope angles of ice-free areas in 100 m elevation bins, differentiated by aspect. Symbols denote the mean elevation and mean slope angle of the studied source rockwalls.**

In the following, we will introduce each study site separately. Note that (i) glacier areas were assigned according to the latest Swiss Glacier Inventory SGI2016 dataset (Linsbauer et al., 2021), (ii) for simplicity we describe only the medial moraines and
associated source rockwalls relevant to this study, and (iii) descriptions below are for the present time, based on the latest datasets available via swisstopo's online map viewer (swisstopo, 2022), unless otherwise noted.

## 2.2 Glacier du Brenay

Brenay (Fig. 1c) is the second largest glacier of the Pigne d'Arolla massif (7.1 km² in 2016; Linsbauer et al., 2021), originating from two basins. Its eastern and main tributary emerges from the southern/southwestern flanks of Pigne d'Arolla, while its



subparallel western tributary originates at the southern flanks of Mont Blanc de Cheilon. Both join at 2900 m elevation, continuing southwest into the Haut Val de Bagnes. In 2020, the glacier was ~5 km long ranging from 2640 to >3600 m elevation. On a relatively snow-free 2017 orthoimage, the snowline is located between 3200 and 3300 m elevation within the western tributary (Fig. 1a,b). Between 1881 and 2020, the glacier continuously retreated, by 1.6 km (GLAMOS, 2021a), and between 1850 and 2010, it lost 30% of its surface area and 60% of its ice volume (Lambiel and Talon, 2019). Between 1934

and 2017, the geodetic glacier-wide mass balance decreased from -0.2 to -0.6 m water equivalent (w.e.) yr$^{-1}$ (GLAMOS, 2021c). The medial moraine of Brenay originates below 3100 m elevation within the western tributary as a lateral moraine and continues for another ~2 km as a medial moraine from the confluence to the glacier terminus. It is sourced from the rockwalls of La Serpentine (3700 m), a ~2 km long mountain ridge that separates the two tributaries. These rockwalls tower up to 700 m above the glacier surface and are mainly composed of quartzdiorite. Whereas rockwalls to the southeast and main tributary are

still largely covered by ice, the southern to northwestern rockwalls are generally ice free. At present, debris deposition occurs both in the accumulation and ablation zone.

### 2.3 Glacier de Cheilon

Cheilon (Fig. 1d) is the third largest glacier of the Pigne d'Arolla massif (3.5 km$^2$ in 2016; Linsbauer et al., 2021). An eastern and a western tributary originate both at the northern flanks of Mont Blanc de Cheilon, join at 2900 m elevation and flow

northwards into the Val des Dix. In 2020, the glacier was ~3 km long, ranging from 2700 to 3500 m elevation. Between 1924 and 2020, it has been shrinking in a step-wise manner, with its terminus retreating 1.2 km (GLAMOS, 2021a). The medial moraine of Cheilon originates within the eastern tributary and extends for ~2 km towards the glacier terminus. It is nourished from three bedrock knobs that emerge between flanking sectors of an icefall in the centre of the eastern tributary at 3000 m elevation. These rockwalls face north, reach heights of up to 300 m, show remnants of ice cover on their flatter tops, and

comprise mainly quartzdiorite. At present, they deposit debris in the glacier ablation zone.

### 2.4 Glacier d'Otemma

The study site at Otemma (Fig. 1e) has been described in detail in Wetterauer et al. (2022a) and we provide only a brief summary here. Otemma is the largest of the five glaciers (12.6 km$^2$ in 2016; Linsbauer et al., 2021) and originates at the southern/southeastern flanks of the Pigne d'Arolla, flowing towards the southwest into the Haut Val de Bagnes. In 2020, its

main trunk was ~6 km long, extending from 2500 to 3000 m elevation. The ELA was located at 3165 m elevation (GLAMOS, 2021b), indicating that the glacier lost most of its former accumulation basin (Fig. 1b). Between 1881 and 2020, the glacier continuously retreated, by 2.5 km (GLAMOS, 2021a), and between 1850 and 2010, it lost 40% of its surface area and 60% of its ice volume (Lambiel and Talon, 2019). Between 1934 and 2017, the geodetic glacier-wide mass balance decreased from -0.4 to -1.3 m w.e. yr$^{-1}$ (GLAMOS, 2021c). Two parallel running medial moraines exist below 3000 m elevation. We refer to them

as the upper (UM) and lower (LM) medial moraine of Otemma, and they can be traced downglacier for >4 and ~3 km distance, respectively. Both are nourished from adjacent but different rock faces of the isolated nunatak Petit Mont Collon at the glacier



head. At present, its northwest-facing rockwalls deliver debris to the UM and are largely ice-fee. Its southwest-facing rockwalls supply the LM and are topped by remnants of a small, nowadays isolated glacier. Overall, these rockwalls are up to 500 m high, mainly composed of orthogneiss and schist, and deposit debris in the ablation zone.

### 2.5 Glacier de Pièce

Pièce (Fig. 1f) is the smallest glacier within the massif (1.3 km$^2$ in 2016; Linsbauer et al., 2021). It originates on the east to northeastern flanks of the Pigne d'Arolla, and its main trunk flows northwards into the Val d'Arolla. In 2020, the main trunk was ~1.5 km long ranging from 2700 to >3100 m elevation. We found no monitoring datasets of the glacier, but historical topographic maps (swisstopo, 2022) indicate its retreat since the end of the 19th century. The medial moraine of Pièce is comparatively short and can be traced for ~1 km downglacier, starting on the easternmost flanks of the Pigne d'Arolla. These

rockwalls face northeast, are up to 300 m high, and mainly composed of orthogneiss and granodiorites. They are largely ice-free, but at their easternmost margin, ice from the southern faces of Pigne d'Arolla reroutes northward into the main trunk. Debris is deposited at 3000 m elevation and currently transported supraglacially.

### 2.6 Glacier de Tsijiore Nouve

Tsijiore Nouve (Fig. 1g) is the second smallest of the five glaciers (2.8 km$^2$ in 2016; Linsbauer et al., 2021). It originates on the northern flanks of the Pigne d'Arolla and takes a northeastward turn into the Val d'Arolla. In 2020, the glacier was ~5 km long, covering a large elevation range from 2300 to >3700 m. The glacier itself can be split into different sections (e.g., Small and Clark, 1974; Small et al., 1979; Small and Gomez, 1981): (i) an upper accumulation basin that spans the upper ~500 m elevation and largely comprises partly crevassed clean ice, (ii) the steep and heavily crevassed Pigne d'Arolla icefall that drops

across the middle ~600 m elevation and hosts the recent ELA, and (iii) a lower and at present heavily debris-covered part in the ablation zone that spans the lower ~300 m elevation and is flanked by large lateral moraines. The glacier's recession history has been discontinuous. Between 1880 and 2020, its terminus retreated by 0.4 km in total, however, disrupted by two major periods of re-advance from 1914 to 1921 and 1971 to 1990 (GLAMOS, 2021a). The sampled medial moraine of Tsijiore Nouve is one of two prominent moraines that both extend along the entire lower part, occupying almost the full recent glacier width.

They are separated from each other by an ice septum of a few tens of meters width and distinguishable by their ridge-like shape. In this study, we focused on the western medial moraine. Unlike the other glaciers, there is no direct apparent continuity to its source rockwalls (Small and Clark, 1974), due to the intervening Pigne d'Arolla icefall and the elongated nature of the accumulation basin. Apparent debris source areas (Fig. 1a) are (i) a 200 m high bedrock face of Pigne d'Arolla that crops out to the north, between flanking sectors of a small icefall, (ii) up to 300 m high rockwalls of an adjacent mountain ridge that

flanks the western rim of the accumulation basin and faces east, and (ii) a 200 m sized bedrock patch recently uncovering at the base of the Pigne d'Arolla icefall (Small and Clark, 1974; Small and Gomez, 1981). All three areas comprise mainly quartzdiorites. Where deposited above the Pigne d'Arolla icefall, debris takes englacial pathways.



### 3 Material and methods

#### 3.1 [10]Be-derived rockwall erosion rates from medial moraine debris

Where rockwalls erode in cosmogenic steady state, their "apparent" rockwall erosion rate $E$ (mm yr$^{-1}$) is described by Eq. (1) (Lal, 1991):

$$E = \left( \frac{P_{sp}(0)}{[^{10}Be]_{rockwall}} - \lambda \right) \frac{\Lambda}{\rho} \tag{1}$$

Assuming an initial concentration of zero, $[^{10}Be]_{rockwall}$ is the [10]Be concentration (atoms g$^{-1}$) at the rockwall surface accumulated during exposure to cosmic rays, $P_{sp}(0)$ is the spallogenic surface production rate (atoms g$^{-1}$ yr$^{-1}$), $\lambda$ is the decay constant (yr$^{-1}$)

(here based on a half-life of $1.387 \pm 0.012$ Myr; Chmeleff et al., 2010; Korschinek et al., 2010), $\Lambda$ is the absorption mean free path (g cm$^{-2}$), and $\rho$ is the material density (here 2.65 g cm$^{-3}$). In our study, we consider rockwall erosion to proceed surface-perpendicular, resulting in lateral rockwall retreat. Moreover, we refer to our rockwall erosion rates as "apparent", because the steady state assumption precludes nuclide inheritance and erosion rate changes over time, and implies that the period of rockwall erosion is longer than the nuclide integration time, which, however, may not apply to recently deglaciating rockwalls

(Wetterauer et al., 2022a).

In Eq. (1), $[^{10}Be]_{rockwall}$ is not equal to the actual measured [10]Be concentration ($[^{10}Be]_{measured}$) in the medial moraine. During supraglacial transport from rockwall to sample location, debris continues to accumulate [10]Be ($[^{10}Be]_{transport}$) (Ward and Anderson, 2011; Scherler and Egholm, 2020; Wetterauer et al., 2022a). Therefore, to obtain $[^{10}Be]_{rockwall}$, we need to know

$[^{10}Be]_{transport}$ and subtract it from $[^{10}Be]_{measured}$. As supraglacial exposure times and rockwall erosion rates vary for individual glaciers, $[^{10}Be]_{transport}$ and its importance relative to $[^{10}Be]_{rockwall}$ likely varies, too. Therefore, estimating and comparing apparent rockwall erosion rates in the Pigne d'Arolla massif requires two independent data acquisitions: (i) quantifying $[^{10}Be]_{measured}$ from debris samples and computing rockwall [10]Be production rates (Sect. 3.1.1), and (ii) estimating glacier flow and debris transport time to correct for $[^{10}Be]_{transport}$ and to approximate the time of rockwall erosion (Sect. 3.1.2).

#### 3.1.1 Debris sampling, [10]Be measurements and [10]Be production in rockwalls

In autumn 2019, we collected 24 new debris samples along medial moraine profiles in the Pigne d'Arolla massif (Table 1, Fig. 1a). The new data expand our previously published 15 samples from the Glacier d'Otemma (Wetterauer et al., 2022a,b) by four new glaciers. Each sample comprises a set of clasts from coarse sands to pebbles (~1-30 mm) and was randomly amalgamated over surface areas covering 5-30 m along a medial moraine by the entire moraine width. At each site, sample

numbers are in ascending order downglacier. None of the moraines was heavily intersected by open crevasses, and debris contribution from valley sidewalls or lateral moraines was not observed. At Brenay, five samples (B1-5) were collected every ~350 m along a 1.5 km long profile. The moraine topography is pronounced, doubling from ~15 to 30 m in height and increasing from ~70 to 110 m in width downglacier. At Cheilon, six samples (C1-6) were collected every ~300 m along a 1.5



km long profile. Downglacier, the moraine topography gains relief, with heights increasing from <1 to ~20 m and widths
increasing from ~50 to 130 m. At Pièce, three samples (P1-3) were collected every ~200 m from a 0.4 km long profile, along
which the moraine topography is <1 m and the width remains at ~50 m. At Tsijiore Nouve, ten samples (TN1-10) were
collected every ~100 m from a 1.0 km long profile. Downglacier, the moraine topography varies, with heights between ~10
and 20 m and widths between ~70 to 90 m. At the last three sampling locations, the moraine flanks had distinct ice cliffs. The
15 Otemma samples by Wetterauer et al. (2022a) were collected every ~500 m from two medial moraines profiles: nine samples
220 from the UM (O/UM1-9) along 4.2 km, and six samples from the LM (O/LM1-6) along 2.7 km. Along both medial moraines,
the initial height of ~3 m flattens out in the central part of the glacier and stays low, and the width decreases from ~20 to 6 m.

*In situ*-produced $^{10}$Be separation was performed on grain size fractions of 1-16 mm (B, C, P and TN samples) and 0.125-4 mm
(O/UM and O/LM samples), as described in Wetterauer et al. (2022b), largely following the procedures of von Blanckenburg
225 et al. (2004). All samples were prepared at the Helmholtz Laboratory for the Geochemistry of the Earth Surface (HELGES) at
the GFZ German Research Centre for Geosciences in Potsdam, Germany. $^{10}$Be/$^{9}$Be ratios (Table 1) were measured at the
accelerator mass spectrometer (AMS) at the University of Cologne, Germany (Dewald et al., 2013) relative to standards KN01-
6-2 and KN01-5-3 (nominal $^{10}$Be/$^{9}$Be ratios: $5.35 \times 10^{-13}$ and $6.32 \times 10^{-12}$, respectively). All ratios were converted into
$[^{10}\text{Be}]_{\text{measured}}$ and corrected for co-processed blanks ($^{10}$Be/$^{9}$Be ratios: $2.16 \times 10^{-15}$, $2.55 \times 10^{-16}$, $1.19 \times 10^{-15}$, $1.56 \times 10^{-15}$).
230

Mean $P_{sp}(0)$ and $\Lambda$ per debris source area were computed on a digital elevation model (DEM) for ice-free rockwalls in 1850,
1973 and 2017 (see Sect. 3.2 for details on the DEM and rockwall outlines). To quantify apparent rockwall erosion rates the
mean values for the 1973-exposed rockwalls were used. In brief, mean $P_{sp}(0)$ are based on the CRONUS functions v2.3 (Balco
et al., 2008), the constant spallation scaling model "St" (Lal, 1991; Stone, 2000), and a sea level high latitude production rate
of $4.01 \pm 0.33$ atoms $\text{g}^{-1}$ $\text{yr}^{-1}$ (Borchers et al., 2016). We considered topographic shielding (Dunne et al., 1999) using the
TopoToolbox v2 function "toposhielding" (Schwanghart and Scherler, 2014) and surface area correction by local slope angles.
We neglected temporal variations in production rates (due to young sample ages of $10^{1}$-$10^{2}$ years; Wetterauer et al., 2022a),
production by muons (typically <1% of spallogenic production), and snow cover shielding (as rockwalls remain largely snow-
free throughout snow cover seasons; swisstopo, 2022). Mean $\Lambda$ were corrected for the surface slope dependency of cosmic ray
attenuation in bedrock (Masarik et al., 2000).



**Table 1: Medial moraine debris samples from the Pigne d'Arolla massif, as well as AMS data with measured $^{10}$Be/$^{9}$Be ratios and $^{10}$Be concentrations ([$^{10}$Be]$_{measured}$).**

| Sample | Latitude | Longitude | Elevation | Distance[b] | Qtz mass | AMS $^{10}$Be/$^{9}$Be $\times 10^{-14} \pm 1\sigma$ | | | Blank[c] | [$^{10}$Be]$_{measured}$ $\times 10^3 \pm 1\sigma$ | | |
|---|---|---|---|---|---|---|---|---|---|---|---|---|
| | (°N) | (°E) | (m) | (m) | (g) | | | | | (atoms g$^{-1}$) | | |
| *Glacier du Brenay* | | | | | | | | | | | | |
| B1 | 45.9672 | 7.4181 | 2880 | 0 | 40.95 | 9.02 | ± | 0.45 | blk3 | 23.58 | ± | 1.21 |
| B2 | 45.9648 | 7.4144 | 2843 | 399 | 34.51 | 10.63 | ± | 0.48 | blk3 | 32.97 | ± | 1.56 |
| B3 | 45.9627 | 7.4110 | 2819 | 746 | 33.36 | 10.30 | ± | 0.48 | blk3 | 33.05 | ± | 1.60 |
| B4 | 45.9608 | 7.4074 | 2755 | 1099 | 37.82 | 10.72 | ± | 0.50 | blk3 | 30.39 | ± | 1.47 |
| B5 | 45.9596 | 7.4032 | 2698 | 1454 | 42.53 | 13.14 | ± | 0.57 | blk3 | 33.33 | ± | 1.50 |
| *Glacier de Cheilon* | | | | | | | | | | | | |
| C1 | 46.0023 | 7.4267 | 2882 | 0 | 35.55 | 1.58 | ± | 0.13 | blk3 | 4.56 | ± | 0.43 |
| C2 | 46.0046 | 7.4247 | 2852 | 301 | 38.93 | 1.29 | ± | 0.12 | blk3 | 3.30 | ± | 0.36 |
| C3 | 46.0073 | 7.4246 | 2839 | 598 | 35.29 | 1.96 | ± | 0.17 | blk3 | 5.69 | ± | 0.52 |
| C4 | 46.0099 | 7.4255 | 2799 | 899 | 37.47 | 1.63 | ± | 0.16 | blk3 | 4.36 | ± | 0.47 |
| C5 | 46.0125 | 7.4264 | 2771 | 1199 | 33.15 | 0.96 | ± | 0.13 | blk3 | 2.75 | ± | 0.43 |
| C6 | 46.0152 | 7.4271 | 2739 | 1499 | 33.64 | 1.49 | ± | 0.14 | blk3 | 4.38 | ± | 0.48 |
| *Glacier d'Otemma/ Upper Medial Moraine[a]* | | | | | | | | | | | | |
| O/UM1 | 45.9653 | 7.4657 | 2918 | 0 | 26.78 | 1.47 | ± | 0.15 | blk2 | 5.53 | ± | 0.57 |
| O/UM2 | 45.9617 | 7.4622 | 2882 | 478 | - | - | - | - | - | 7.41 | ± | 0.79 |
| O/UM3 | 45.9584 | 7.4559 | 2837 | 1088 | 26.88 | 1.73 | ± | 0.15 | blk2 | 6.52 | ± | 0.57 |
| O/UM4 | 45.9543 | 7.4504 | 2789 | 1712 | 22.82 | 8.50 | ± | 0.40 | blk2 | 38.32 | ± | 1.86 |
| O/UM5 | 45.9507 | 7.4458 | 2747 | 2248 | 23.06 | 2.47 | ± | 0.17 | blk2 | 10.96 | ± | 0.78 |
| O/UM6 | 45.9483 | 7.4405 | 2707 | 2743 | 22.97 | 3.48 | ± | 0.22 | blk2 | 15.52 | ± | 0.98 |
| O/UM7 | 45.9455 | 7.4352 | 2663 | 3258 | 23.39 | 6.19 | ± | 0.39 | blk1 | 26.34 | ± | 1.74 |
| O/UM8 | 45.9431 | 7.4297 | 2600 | 3758 | 22.39 | 4.43 | ± | 0.26 | blk2 | 20.32 | ± | 1.23 |
| O/UM9 | 45.9407 | 7.4248 | 2547 | 4224 | 22.67 | 4.12 | ± | 0.25 | blk2 | 18.59 | ± | 1.14 |
| *Glacier d'Otemma/ Lower Medial Moraine[a]* | | | | | | | | | | | | |
| O/LM1 | 45.9613 | 7.4621 | 2875 | 0 | - | - | - | - | - | 17.78 | ± | 1.14 |
| O/LM2 | 45.9582 | 7.4560 | 2835 | 581 | 22.64 | 4.94 | ± | 0.30 | blk2 | 22.37 | ± | 1.39 |
| O/LM3 | 45.9542 | 7.4507 | 2791 | 1184 | 14.54 | 2.07 | ± | 0.18 | blk1 | 13.09 | ± | 1.32 |
| O/LM4 | 45.9504 | 7.4459 | 2749 | 1745 | 20.31 | 2.72 | ± | 0.21 | blk1 | 12.66 | ± | 1.10 |
| O/LM5 | 45.9480 | 7.4407 | 2712 | 2228 | 22.90 | 3.28 | ± | 0.22 | blk2 | 14.68 | ± | 0.99 |
| O/LM6 | 45.9450 | 7.4354 | 2668 | 2754 | 17.11 | 5.34 | ± | 0.32 | blk1 | 30.81 | ± | 2.00 |
| *Glacier de Pièce* | | | | | | | | | | | | |
| P1 | 45.9967 | 7.4703 | 2859 | 0 | 40.45 | 1.86 | ± | 0.17 | blk4 | 4.40 | ± | 0.46 |
| P2 | 45.9983 | 7.4692 | 2802 | 203 | 32.19 | 1.50 | ± | 0.16 | blk4 | 4.35 | ± | 0.53 |
| P3 | 46.0001 | 7.4690 | 2775 | 406 | 31.87 | 1.33 | ± | 0.14 | blk4 | 3.83 | ± | 0.49 |
| *Glacier de Tsijiore Nouve* | | | | | | | | | | | | |
| TN1 | 46.0067 | 7.4554 | 2530 | 0 | 39.91 | 1.07 | ± | 0.14 | blk4 | 2.25 | ± | 0.37 |
| TN2 | 46.0075 | 7.4556 | 2526 | 98 | 40.84 | 1.00 | ± | 0.13 | blk4 | 2.05 | ± | 0.33 |
| TN3 | 46.0084 | 7.4558 | 2509 | 199 | 39.70 | 0.82 | ± | 0.11 | blk4 | 1.68 | ± | 0.31 |
| TN4 | 46.0096 | 7.4561 | 2491 | 334 | 40.78 | 0.96 | ± | 0.13 | blk4 | 2.05 | ± | 0.34 |
| TN5 | 46.0105 | 7.4565 | 2480 | 432 | 40.63 | 0.87 | ± | 0.11 | blk4 | 1.84 | ± | 0.31 |
| TN6 | 46.0113 | 7.4568 | 2466 | 532 | 40.64 | 1.61 | ± | 0.15 | blk4 | 3.71 | ± | 0.39 |
| TN7 | 46.0122 | 7.4572 | 2458 | 632 | 40.71 | 1.11 | ± | 0.15 | blk4 | 2.45 | ± | 0.39 |
| TN8 | 46.0133 | 7.4578 | 2457 | 765 | 38.20 | 1.40 | ± | 0.14 | blk4 | 3.40 | ± | 0.41 |
| TN9 | 46.0140 | 7.4586 | 2433 | 865 | 39.72 | 0.90 | ± | 0.13 | blk4 | 1.95 | ± | 0.36 |
| TN10 | 46.0145 | 7.4597 | 2431 | 963 | 40.67 | 1.14 | ± | 0.13 | blk4 | 2.51 | ± | 0.35 |

[a] Samples O/UM1-9 and O/LM1-6 within this study correspond to published samples UM1-9/f and LM1-6/f in Wetterauer et al. (2022a,b).

[b] Downglacier distance from the uppermost sample of the respective medial moraine.

[c] Process blank used to correct respective sample batches, where corresponding AMS $^{10}$Be/$^{9}$Be ratios are blk1 = 2.16×10$^{-15}$, blk2 = 2.55×10$^{-16}$, blk3 = 1.19×10$^{-15}$, blk4 = 1.56×10$^{-15}$.



### 3.1.2 Debris transport time and additional $^{10}$Be production

To assess [$^{10}$Be]$_{transport}$, additionally accumulated during debris transport from source rockwalls to sampling location, we used the simple 1-D debris particle trajectory model developed for Otemma and described in detail in Wetterauer et al. (2022a,b). For every glacier, we estimated the horizontal and vertical ice velocities along the sampled medial moraine through time, using observed surface velocities and mass balance-dependent particle burial/re-emergence, respectively. For a given time span, we then fed the glacial conveyor belt with particles and tracked their downglacier and englacial/supraglacial position. For each

sample, we estimated its sample age (i.e., time of debris erosion) and [$^{10}$Be]$_{transport}$ from the modelled transport duration, by averaging the trajectories of those particles that arrived supraglacially within ± 30 m of the sample location in 2019.

We determined glacier surface velocities and elevations and a generalized ELA history for the four new study sites Brenay, Cheilon, Pièce and Tsijiore Nouve. Unfortunately, recent surface velocities obtained from cross-correlation of satellite imagery

tend to be poorly resolved for the relatively small and slow glaciers of the Pigne d'Arolla massif. Especially close to the rockwalls, uncertainties are typically similar to the derived velocities (Fig. 3b,d,f,h; Millan et al., 2022). Therefore, we estimated horizontal glacier surface velocities by manually tracing the displacement of medial moraine boulders across orthoimages of the last ~40 years (swisstopo, 2022). Boulder displacements were measured between successive othoimages from the years 1977, 1983, 1988, 1995, 1999, 2005, 2007, 2010, 2013, 2017, and 2020 (Fig. 3a,c,e,g). We considered boulder

displacements parallel to profiles of the respective medial moraine and averaged boulder velocities over the respective time period between two orthoimages. Although exact acquisition dates were unavailable, the orthoimages presumably stem from a similar time during the summer season, as the snowline was located high, and the medial moraines were largely snow-free, except during the period 1995-1999. The positional accuracy for the orthoimages increases from 1 m before 1999 to 0.1 m at present (swisstopo, 2022). Due to the increasing orthoimage resolution, we were able to identify and track more boulders

during more recent periods, which allowed us to obtain continuous downglacier velocity profiles. For earlier periods, this was hampered by fewer velocity estimates with greater uncertainties. To nevertheless obtain temporally continuous velocity profiles, we fitted the most recent tracking period 2017-2020, using SLM tools (D'Errico, 2022), and shifted this fit vertically to the higher velocities during earlier periods using least squares (Fig. 3b,d,f,h). Velocity changes over the entire period 1977-2020 were obtained by linear interpolation between the individual periods (Fig. 4b). For the years 1880-1977, velocity changes

were approximated by aligning the gradient of velocity change with the linear gradient of the respective glacier's length change for the same period, obtained from long-term glacier length monitoring records (Fig. 4, GLAMOS, 2021a).



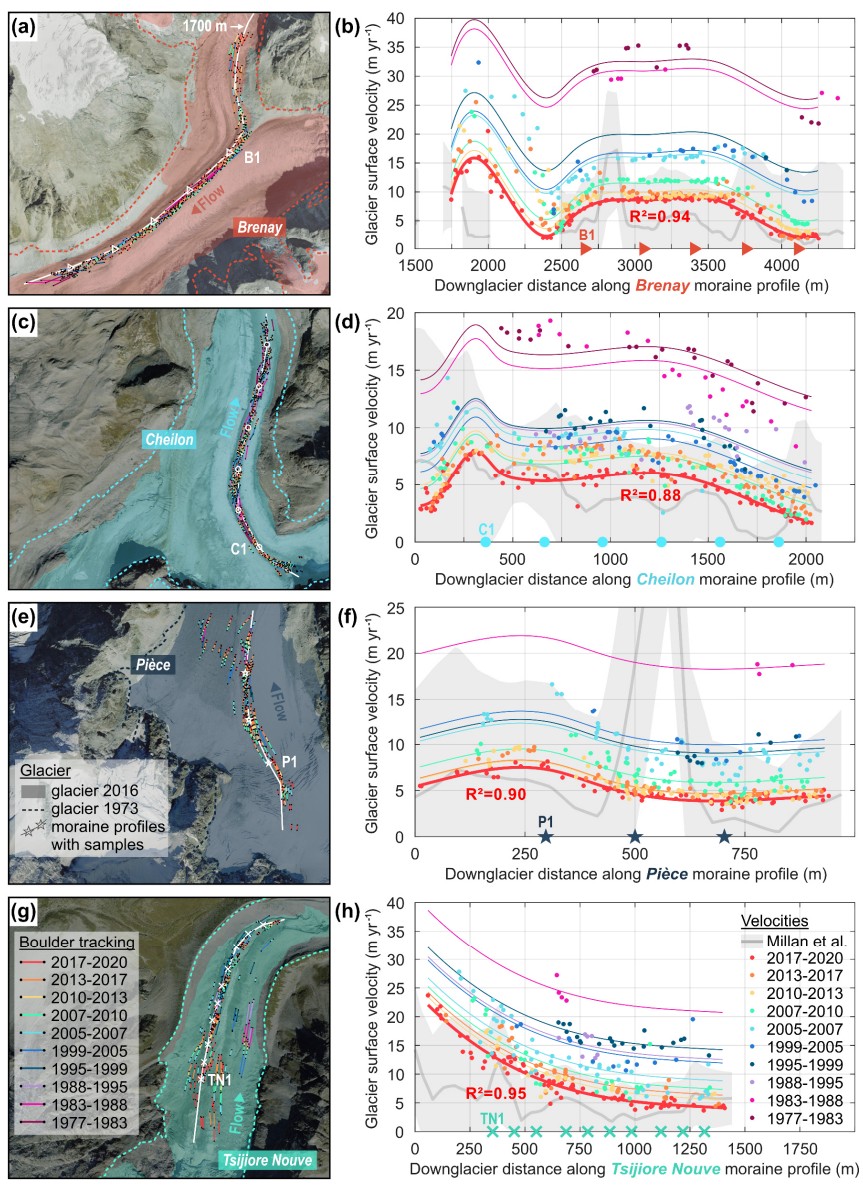

**Figure 3: Glacier surface velocities along the four newly sampled medial moraines, reconstructed from 10 boulder tracking periods between the years 1977 to 2020. (a, c, e, g) Tracked boulder displacements along downglacier profiles, which trace the sampled medial moraines. Where a boulder could be followed across several tracking periods color-coded lines are connected. Note that**






boulder displacements at Brenay could only be tracked after a profile distance of ~1700 m, due to englacial transport along the first half of the profile (2017 orthoimage by swisstopo, 2022; glacier extents by Linsbauer et al., 2021, Müller et al., 1976). (b, d, f, h) Glacier surface velocities through time derived from the mapped boulder tracks. The solid red line and $R^2$ indicate the fit of the velocities obtained for the most recent tracking period 2017-2020, thin coloured lines reflect the same fit applied to the nine older tracking periods. For reference, remotely sensed velocities ± uncertainties by Millan et al. (2022) are shown in grey in the background, and downglacier sample locations are indicated on the x-axis. All datasets indicate a systematic slowdown of glacier flow towards the present. For reconstructions of Glacier d'Otemma see Wetterauer et al. (2022b).

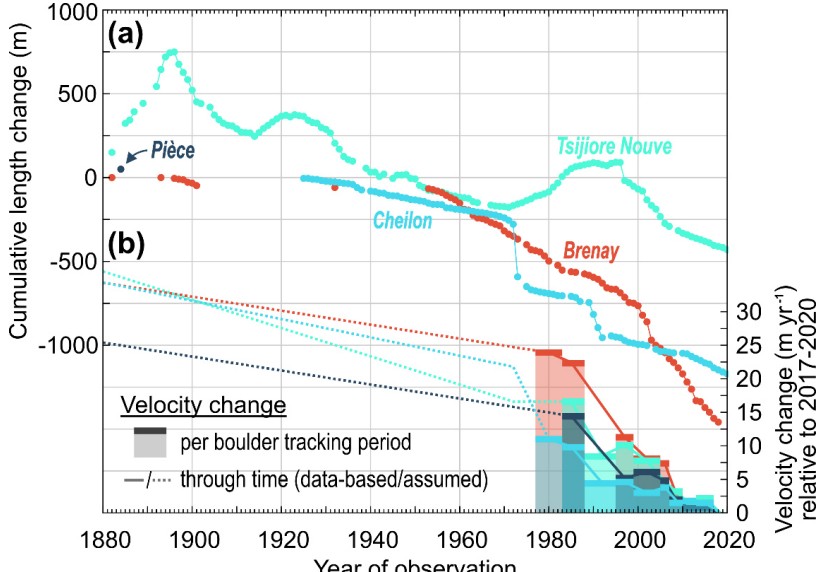

Figure 4: Glacier change at the four new study sites since 1880. (a) Long-term monitoring records of glacier length change (GLAMOS, 2021a). Note that markers from consecutive years are connected and that no length change record exists for Glacier de Pièce. (b) Reconstructed temporal changes in glacier surface velocity, relative to the most recent boulder tracking period 2017-2020. Per boulder tracking period, the velocity change corresponds to the y-axis shift of its fit line to the solid red fit line in Fig. 3b,d,f,h. On a year-to-year basis, velocity changes are based on linear interpolation between the velocity changes of the different tracking periods. Pre-1977 changes in velocities were approximated with reference to the linear gradient of the glacier's length change in (a). Due to lacking length data at Glacier de Pièce, pre-1977 assumptions there align with Glacier du Brenay. For reconstructions of Glacier d'Otemma see Wetterauer et al. (2022b).

To constrain ice surface lowering along the four new moraine profiles through time, we interpolated between contour lines for the year 1880 extracted from the Siegfried Map (first edition 1870-1926), and a topographic map from 2016 (swisstopo, 2022), using a three step linear decline (Wetterauer et al., 2022b). ELA elevations through time were adopted from a recent ELA reconstruction dataset for this region (Žebre et al., 2021). Model settings of a generalized vertical mass balance profile (gradient: 0.7 m w.e. yr[-1] per 100 m elevation, maximum snow accumulation above the ELA: 0.75 m w.e. yr[-1]) and the computation of $^{10}$Be production rates as a function of transport time, downglacier distance and burial depth were adopted from Wetterauer et al. (2022b). At Brenay, where the elongated source rockwalls of La Serpentine lead to debris deposition along the first 2.3 km of the profile (Fig. 1), we additionally considered the mixing of debris particles of different age and $[^{10}Be]_{transport}$





at a sample location. There, particle trajectories were modelled from various source locations and weighted according to the above rockwall area, which we defined by flow directions using the steepest decent approach (Fig. S2). At the other sites, where debris deposition occurs approximately at the profile's head, trajectories were modelled from the profile head as single point source.

We acknowledge that our reconstruction of glacier surface velocities and elevations contains several sources of unquantified uncertainties. However, properly modelling these glaciers and the particle transport (e.g., Scherler and Egholm, 2020), requires even more empirical constraints that are currently unavailable, and, as we will show in Sect. 5.2, our findings are largely insensitive to the $[^{10}Be]_{transport}$.

**3.2 Source rockwall analysis**

To compare the different medial moraine debris source areas in the Pigne d'Arolla massif and quantitatively assess the extent to which they are affected by deglaciation, we obtained source rockwall outlines for a recent (2017) and two past time slices (1973, 1850). Recent outlines were defined manually by mapping the ice-free rockwalls on a 2017 orthoimage (swisstopo, 2022). Former outlines of ice-free rockwalls are based on reconstructed glacier outlines from the years 1973 and 1850 (Müller et al., 1976; modified from Maisch et al., 2000). Across the defined rockwalls (Fig. S1), area, elevation, slope and aspect were
determined for each pixel in a 30 m resolution digital elevation model (DEM; global raster dataset SRTM GL1; NASA Shuttle Radar Topography Mission SRTM, 2013). The surface area covered per pixel was corrected for its local slope angle. To visualize changes in ice cover across the source rockwalls, we further compiled historical photographs from the online archives of swisstopo (2022) and the ETH Library (2022).

**4 Results**

Below, the four new datasets of Brenay, Cheilon, Pièce and Tsijiore Nouve are presented in summary form and in context with the two published Otemma datasets (Wetterauer et al., 2022a,b). Details on $[^{10}Be]$ measurement results (Table 1), model-based corrections and derived apparent rockwall erosion rates (Table 2) of individual medial moraine samples as well as on distinct debris source area properties (Table 3) are provided in the data tables as indicated.

**4.1 Glacier surface velocities, estimated $^{10}Be$-transport concentrations, and sample ages**

The tracked boulder displacements resulted in well-defined glacier surface velocities along our four new sampled medial moraines for the most recent period (Fig. 3b,d,f,h). The further back in time, the more fragmented the records are. Our approach to fit the most recent tracking period and shift the velocity fit vertically, appears to yield reasonable fits to velocity estimates from earlier tracking periods. However, observations prior to 1995 are scarce, especially for Piece and Tsijiore Nouve, and the shifted fit does not agree very well with the pre-1988 estimates for Cheilon.




**Table 2: Model results of downglacier debris transport with samples ages, burial depth and additional $^{10}$Be accumulation ($[^{10}$Be$]_{transport}$), as well as transport-corrected $^{10}$Be concentrations ($[^{10}$Be$]_{rockwall}$) and derived minimum and maximum apparent rockwall erosion rates.**

| Sample | Age[b] | Burial[b,c] / max | $[^{10}$Be$]_{transport}$[b] ×10³ | $[^{10}$Be$]_{rockwall}$[d] ×10³ | | | Apparent rockwall erosion rate[e,f] min (uncorr.) | | | max (transport-corr.) | | |
|---|---|---|---|---|---|---|---|---|---|---|---|---|
| | (yrs) | (m) | (atoms g⁻¹) | (atoms g⁻¹) | | ± 1σ | (mm yr⁻¹) | | ± 1σ | (mm yr⁻¹) | | ± 1σ |
| *Glacier du Brenay* | | | | | | | | | | | | |
| B1 | 52 | -13.4 | 0.7 | 22.9 | ± | 1.2 | 1.0 | ± | 0.1 | 1.0 | ± | 0.1 |
| B2 | 63 | -13.5 | 0.9 | 32.1 | ± | 1.6 | 0.7 | ± | 0.1 | 0.7 | ± | 0.1 |
| B3 | 72 | -10.2 | 1.1 | 32.0 | ± | 1.6 | 0.7 | ± | 0.1 | 0.7 | ± | 0.1 |
| B4 | 83 | -7.8 | 1.5 | 28.8 | ± | 1.5 | 0.8 | ± | 0.1 | 0.8 | ± | 0.1 |
| B5 | 96 | -9.9 | 1.9 | 31.5 | ± | 1.5 | 0.7 | ± | 0.1 | 0.7 | ± | 0.1 |
| *Glacier de Cheilon* | | | | | | | | | | | | |
| C1 | 36 | -0.2 | 1.0 | 3.5 | ± | 0.4 | 3.9 | ± | 0.5 | 5.1 | ± | 0.7 |
| C2 | 50 | -1.8 | 1.3 | 2.0 | ± | 0.4 | 5.5 | ± | 0.8 | 8.9 | ± | 1.8 |
| C3 | 61 | -0.2 | 1.7 | 4.0 | ± | 0.5 | 3.2 | ± | 0.4 | 4.5 | ± | 0.7 |
| C4 | 70 | 0.0 | 2.0 | 2.4 | ± | 0.5 | 4.1 | ± | 0.6 | 7.6 | ± | 1.6 |
| C5 | 80 | 0.0 | 2.3 | 0.5 | ± | 0.4 | 6.6 | ± | 1.2 | 36.7[g] | ± | 32.3[g] |
| C6 | 92 | 0.0 | 2.6 | 1.8 | ± | 0.5 | 4.1 | ± | 0.6 | 10.0 | ± | 2.8 |
| *Glacier d'Otemma/ Upper Medial Moraine[a]* | | | | | | | | | | | | |
| O/UM1 | 45 | 0.0 | 1.4 | 4.2 | ± | 0.6 | 3.8 | ± | 0.5 | 5.1 | ± | 0.8 |
| O/UM2 | 77 | 0.0 | 2.4 | 5.0 | ± | 0.8 | 2.9 | ± | 0.4 | 4.2 | ± | 0.7 |
| O/UM3 | 103 | -1.2 | 2.8 | 3.7 | ± | 0.6 | 3.3 | ± | 0.4 | 5.8 | ± | 1.0 |
| O/UM4 | 130 | -5.0 | 2.5 | 35.8 | ± | 1.9 | 0.6 | ± | 0.1 | 0.6 | ± | 0.1 |
| O/UM5 | 148 | -9.1 | 2.3 | 8.6 | ± | 0.8 | 1.9 | ± | 0.2 | 2.5 | ± | 0.3 |
| O/UM6 | 167 | -12.4 | 2.3 | 13.2 | ± | 1.0 | 1.4 | ± | 0.1 | 1.6 | ± | 0.2 |
| O/UM7 | 182 | -14.7 | 2.4 | 24.0 | ± | 1.7 | 0.8 | ± | 0.1 | 0.9 | ± | 0.1 |
| O/UM8 | 199 | -16.2 | 2.4 | 17.9 | ± | 1.2 | 1.0 | ± | 0.1 | 1.2 | ± | 0.1 |
| O/UM9 | 206 | -16.7 | 2.5 | 16.1 | ± | 1.1 | 1.1 | ± | 0.1 | 1.3 | ± | 0.1 |
| *Glacier d'Otemma/ Lower Medial Moraine[a]* | | | | | | | | | | | | |
| O/LM1 | 35 | 0.0 | 1.0 | 16.7 | ± | 1.1 | 1.2 | ± | 0.1 | 1.3 | ± | 0.1 |
| O/LM2 | 67 | 0.0 | 2.0 | 20.3 | ± | 1.4 | 1.0 | ± | 0.1 | 1.1 | ± | 0.1 |
| O/LM3 | 94 | 0.0 | 2.8 | 10.3 | ± | 1.3 | 1.7 | ± | 0.2 | 2.2 | ± | 0.3 |
| O/LM4 | 115 | -0.2 | 3.4 | 9.2 | ± | 1.1 | 1.7 | ± | 0.2 | 2.4 | ± | 0.3 |
| O/LM5 | 134 | -1.8 | 3.5 | 11.2 | ± | 1.0 | 1.5 | ± | 0.2 | 2.0 | ± | 0.2 |
| O/LM6 | 152 | -4.2 | 3.3 | 27.6 | ± | 2.0 | 0.7 | ± | 0.1 | 0.8 | ± | 0.1 |
| *Glacier de Pièce* | | | | | | | | | | | | |
| P1 | 27 | 0.0 | 0.8 | 3.6 | ± | 0.5 | 4.5 | ± | 0.6 | 5.5 | ± | 0.8 |
| P2 | 39 | -1.1 | 1.0 | 3.3 | ± | 0.5 | 4.5 | ± | 0.7 | 5.9 | ± | 1.1 |
| P3 | 50 | -1.6 | 1.2 | 2.6 | ± | 0.5 | 5.1 | ± | 0.8 | 7.6 | ± | 1.6 |
| *Glacier de Tsijiore Nouve* | | | | | | | | | | | | |
| TN1 | 16 | 0.0 | 0.4 | 1.9 | ± | 0.4 | 8.8 | ± | 1.6 | 10.6 | ± | 2.3 |
| TN2 | 21 | 0.0 | 0.5 | 1.6 | ± | 0.3 | 9.7 | ± | 1.7 | 12.7 | ± | 2.9 |
| TN3 | 27 | 0.0 | 0.6 | 1.1 | ± | 0.3 | 11.8 | ± | 2.4 | 18.7 | ± | 5.8 |
| TN4 | 34 | 0.0 | 0.8 | 1.3 | ± | 0.3 | 9.7 | ± | 1.8 | 15.6 | ± | 4.4 |
| TN5 | 40 | 0.0 | 0.9 | 0.9 | ± | 0.3 | 10.8 | ± | 2.0 | 21.3 | ± | 7.3 |
| TN6 | 45 | 0.0 | 1.0 | 2.7 | ± | 0.4 | 5.3 | ± | 0.7 | 7.4 | ± | 1.2 |
| TN7 | 50 | 0.0 | 1.1 | 1.3 | ± | 0.4 | 8.1 | ± | 1.5 | 15.1 | ± | 4.7 |
| TN8 | 56 | 0.0 | 1.3 | 2.1 | ± | 0.4 | 5.8 | ± | 0.9 | 9.3 | ± | 1.9 |
| TN9 | 61 | 0.0 | 1.4 | 0.6 | ± | 0.4 | 10.2 | ± | 2.1 | 34.5 | ± | 21.7 |
| TN10 | 65 | 0.0 | 1.5 | 1.1 | ± | 0.3 | 7.9 | ± | 1.3 | 18.9 | ± | 6.5 |

[a] Samples O/UM1-9 and O/LM1-6 within this study correspond to published samples UM1-9/f and LM1-6/f in Wetterauer et al. (2022a,b).

[b] Computed using a simple 1-D debris particle trajectory model (Wetterauer et al., 2022b).



<sup>c</sup> Maximum burial depth modelled for debris particles of a sample. Negative numbers indicate partially englacial transport, 0 indicates exclusively supraglacial transport.

<sup>d</sup> Calculated by subtracting $[^{10}Be]_{transport}$ from $[^{10}Be]_{measured}$. 1σ estimates correspond to analytical uncertainties of $[^{10}Be]_{measured}$.

<sup>e</sup> Calculated using the mean spallogenic production rates and absorption mean free paths of the respective 1973-exposed source rockwalls listed in Table 3. 1σ estimates are based on the analytical uncertainties of $[^{10}Be]_{measured}$.

<sup>f</sup> Minimum rates were derived from $[^{10}Be]_{measured}$, uncorrected for downglacier transport. Maximum rates were derived from $[^{10}Be]_{rockwall}$, corrected for downglacier transport.

<sup>g</sup> Value excluded from any erosion rate-based analysis due to unreasonable $[^{10}Be]_{rockwall}$ estimate (see Sect. 5.1).

Based on the modelled debris trajectories at Brenay, Cheilon, Pièce, and Tsijiore Nouve, transport occurred supraglacial over most of the downglacier distance, in part even exclusively (Fig. S3-S4). The resulting $[^{10}Be]_{transport}$ reach values up to $3\times10^3$

atoms g$^{-1}$ (Table 2). The estimated sample ages overlap and are generally younger than 100 years (Table 2, Fig. 5). In contrast, sample ages at Otemma UM and LM cover the last 200 and 150 years, respectively (Wetterauer et al., 2022a).

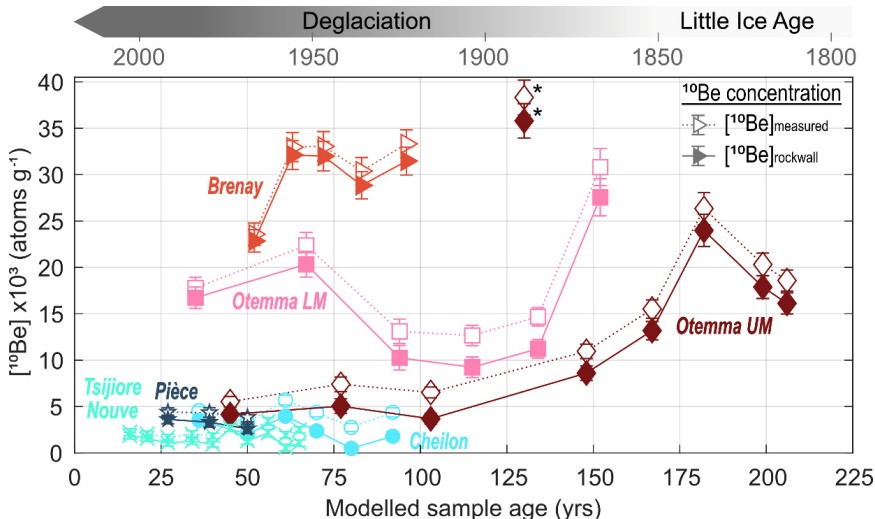

**Figure 5: Comparison of uncorrected ($[^{10}Be]_{measured}$) and transport-corrected ($[^{10}Be]_{rockwall}$) medial moraine $[^{10}Be]$ records of the Pigne d'Arolla massif through time (data from Glacier d'Otemma by Wetterauer et al., 2022a). Concentrations (±1σ analytical**
**error) are plotted against modelled sample ages, reflecting the timing of rockwall erosion either during the Little Ice Age or the following deglaciation period, as indicate by the timeline above. Note that a re-measurement of a high concentration outlier at Otemma UM (\*) at a slightly coarser grain size fraction fits the temporal trend well (see Fig. 6a in Wetterauer et al., 2022a).**

## 4.2 Measured $^{10}$Be and estimated $^{10}$Be-rockwall concentrations

The $[^{10}Be]_{measured}$ of our 24 new medial moraine debris samples range between $2\times10^3$ and $33\times10^3$ atoms g$^{-1}$ (Table 1, Fig. 5).
At Brenay, Cheilon, Pièce, and Tsijiore Nouve, $[^{10}Be]_{measured}$ are rather uniform along the medial moraine averaging at $31\times10^3$, $4\times10^3$, $4\times10^3$, and $2\times10^3$ atoms g$^{-1}$, respectively. Correcting $[^{10}Be]_{measured}$ for $[^{10}Be]_{transport}$ results in overall lower $[^{10}Be]_{rockwall}$ (Table 2, Fig. 5). At Brenay, where $[^{10}Be]_{measured}$ is high, $[^{10}Be]_{transport}$ is only a small fraction of the concentration (3-6%). For samples with low $[^{10}Be]_{measured}$ at Cheilon, Pièce, and Tsijiore Nouve, the correction can account for up to half (17-49%), and





in individual cases even more (58-82%). Still, the general tendency of the individual [$^{10}$Be]$_{measured}$ records towards temporal
consistency is maintained in [$^{10}$Be]$_{rockwall}$. The concentration ranges are comparable to the Otemma datasets (Fig. 5), however,
medial moraine [$^{10}$Be] at Otemma were found to be more variable and to systematically increase downglacier at Otemma UM
(Wetterauer et al., 2022a).

### 4.3 Apparent rockwall erosion rates

We provide two estimates of apparent rockwall erosion rates: based on the uncorrected [$^{10}$Be]$_{measured}$ and on the transport-
corrected [$^{10}$Be]$_{rockwall}$ (Table 2). At Brenay, the mean uncorrected/corrected rate estimates ($\pm 1\sigma$) both round to $0.8 \pm 0.1$ mm
yr$^{-1}$. For Cheilon and Pièce, mean uncorrected/corrected rate estimates deviate more strongly and are comparatively higher
with $4.6 \pm 1.2 / 7.2 \pm 2.4$ and $4.7 \pm 0.4 / 6.3 \pm 1.1$ mm yr$^{-1}$, respectively. Mean uncorrected/corrected rate estimates are much
higher at Tsijiore Nouve with $8.8 \pm 2.1 / 16.4 \pm 7.8$ mm yr$^{-1}$, which we consider to be unreasonable for reasons discussed in
Sect. 5.1. For comparison, mean uncorrected/corrected rate estimates at Otemma UM and LM are $1.9 \pm 1.2 / 2.6 \pm 2.0$ and $1.3$
$\pm 0.4 / 1.6 \pm 0.6$ mm yr$^{-1}$, respectively (Wetterauer et al., 2022a). We note that our erosion rate analyses are based on the ice-
free rockwall areas of 1973, because debris deposition, respectively rockwall erosion, presumably occurred largely between
1850 and 1973 (Fig. 5).

### 4.4 Source rockwall morphology

We compare the debris source areas in the Pigne d'Arolla massif by area, aspect and mean slope of the ice-free rockwall areas
in the years 1850, 1973 and 2017 (Table 3). Overall, rockwalls are located between 3100 and 3400 m elevation. While the
rockwalls at Brenay represent by far the largest debris source area, the other rockwalls are comparatively small. In 1973,
southwest-facing rockwalls at Brenay were inclined by 43° on average. At the northwest- to east-facing rockwalls of Cheilon,
Pièce and Tsijiore Nouve mean slope angles were steeper, ranging between 48 and 51°. These observations are consistent with
mean slope angles at Otemma, being in 1973 steeper at the northwest-facing rockwalls (43°) of UM than at the southwest-
facing rockwalls (36°) of LM (Wetterauer et al., 2022a).

Source rockwall deglaciation in the Pigne d'Arolla massif after 1850 differs locally. Between 1850 and 1973, ice-free rockwalls
at Brenay, Cheilon, Otemma UM, Pièce and Tsijiore Nouve expanded by <5%, resulting in minimal changes in mean rockwall
elevation. At Otemma LM, however, ice-free rockwalls expanded by ~150%, and the mean elevation markedly dropped.
Between 1973 and 2017, Brenay, Pièce and Tsijiore Nouve rockwalls seem to have expanded by only ~20%. Thus, shifts in
mean elevations have remained again small, except at Tsijiore Nouve, where recent bedrock exposure at the Pigne d'Arolla
icefall reduced the mean rockwall elevation. In contrast, ice-free areas at Cheilon and Otemma UM and LM rockwalls
approximately doubled since 1973 by expanding with 86 to 129%. Whereas at Cheilon mean rockwall elevation has shifted
upward, Otemma rockwalls expanded predominantly downward. Visually, the degree of rockwall deglaciation is also evident
on historical photographs (Fig. 6), particularly the significant ice cover changes across Otemma LM rockwalls.





**Table 3: Geomorphic parameters of source rockwalls in the years 1850, 1973 and 2017, as well as [10]Be production rates and absorption mean free paths.**

| Rockwalls | Lithology[a] | Area[b] (km²) | ± Δ[c] | Elevation / mean (m) | ± Δ[d] | Slope / mean (°) | Aspect / mean (°) | | P$_{sp}$(0) / mean (atoms g⁻¹ yr⁻¹) | Λ / mean (g cm⁻²) |
|---|---|---|---|---|---|---|---|---|---|---|
| *Glacier du Brenay* | | | | | | | | | | |
| 1850 | QD | 1.26 | 1% | 3387 | -2 m | 43 | 238 | SW | 43 | 141 |
| 1973 | QD | 1.27 | | 3385 | | 43 | 238 | SW | 43 | 141 |
| 2017 | QD | 1.53 | 20% | 3374 | -11 m | 41 | 230 | SW | 44 | 142 |
| *Glacier de Cheilon* | | | | | | | | | | |
| 1850 | QD | 0.08 | 3% | 3131 | 7 m | 52 | 317 | NW | 35 | 136 |
| 1973 | QD | 0.08 | | 3138 | | 51 | 316 | NW | 35 | 137 |
| 2017 | QD | 0.15 | 86% | 3181 | 43 m | 42 | 311 | NW | 38 | 141 |
| *Glacier d'Otemma/ Upper Medial Moraine* | | | | | | | | | | |
| 1850 | OG + S | 0.10 | 0% | 3174 | 0 m | 43 | 293 | NW | 39 | 143 |
| 1973 | OG + S | 0.10 | | 3174 | | 43 | 293 | NW | 39 | 143 |
| 2017 | OG + S | 0.23 | 129% | 3131 | -43 m | 39 | 292 | W | 38 | 144 |
| *Glacier d'Otemma/ Lower Medial Moraine* | | | | | | | | | | |
| 1850 | OG + S | 0.11 | 151% | 3283 | -121 m | 41 | 214 | SW | 42 | 142 |
| 1973 | OG + S | 0.27 | | 3162 | | 36 | 229 | SW | 40 | 145 |
| 2017 | OG + S | 0.51 | 87% | 3126 | -36 m | 31 | 236 | SW | 40 | 146 |
| *Glacier de Pièce* | | | | | | | | | | |
| 1850 | OG + GD | 0.08 | 0% | 3175 | 0 m | 49 | 26 | NE | 37 | 140 |
| 1973 | OG + GD | 0.08 | | 3175 | | 49 | 26 | NE | 37 | 140 |
| 2017 | OG + GD | 0.10 | 16% | 3177 | 2 m | 48 | 27 | NE | 38 | 140 |
| *Glacier de Tsijiore Nouve* | | | | | | | | | | |
| 1850 | QD | 0.21 | 0% | 3339 | 0 m | 48 | 90 | E | 39 | 136 |
| 1973 | QD | 0.21 | | 3339 | | 48 | 90 | E | 39 | 136 |
| 2017 | QD | 0.26 | 23% | 3247 | -92 m | 47 | 78 | E | 38 | 137 |

[a] Crystalline rocks of the Arolla series, where GD = Granodiorite, OG = Orthogneiss, QD = Quartzdiorite, S = Schist.

[b] Corrected for slope.

[c] Area expansion (%) between 1850 and 1973 and between 1973 and 2017, respectively.

[d] Mean elevation gain/loss (m) between 1850 and 1973 and between 1973 and 2017, respectively, indicating predominant direction of area expansion.



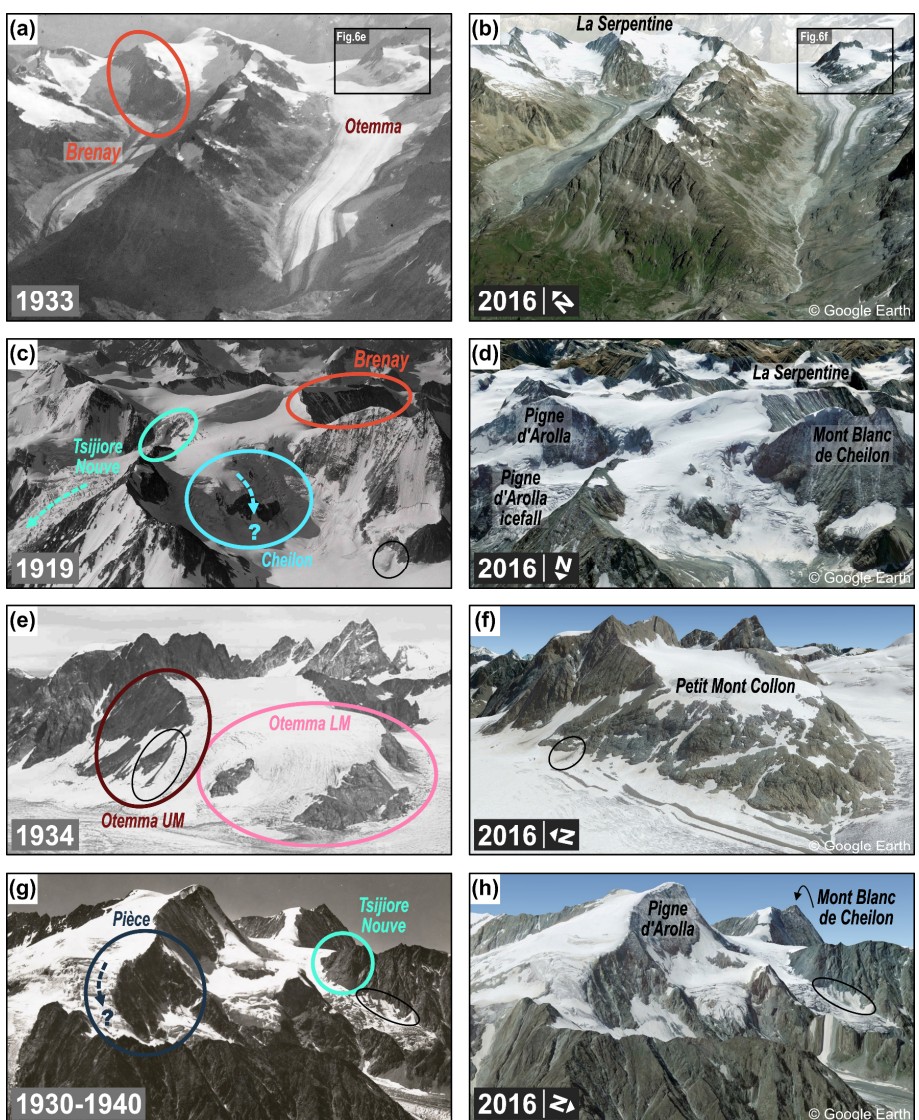

**Figure 6:** Ice cover changes across deglaciating source rockwalls in the Pigne d'Arolla massif between the beginning of the (a, c, e, g) 20th and (b, d, f, h) 21st century. For each historical photograph on the left, the approximately same view from Google Earth (https://earth.google.com) is shown on the right. The years shown are indicated (historical photograghs by ETH Library, 2022, swisstopo, 2022). For clarity, debris source areas are only indicated on the left (coloured circles), and mountain peaks or ridges mentioned in the text are only labelled on the right. Areas where subglacial erosion may bias our medial moraine records are indicated (dashed arrows). Note the examples of small-scale rock falls representative for rockwall erosion (black circles).



## 5 Discussion

### 5.1 How "apparent" are the rockwall erosion rate estimates?

The medial moraine records of the Pigne d'Arolla massif - including the Otemma datasets (Wetterauer et al., 2022a) - cover a wide range of apparent rockwall erosion rates (Table 2), largely spanning between 0.6 and 10.0 mm yr$^{-1}$, and presumably
covering the last 200 years. Our particle trajectory modelling suggests that our new samples from Brenay, Cheilon, Pièce, and Tsijiore Nouve cover the post-LIA deglaciation period and their [$^{10}$Be] indicate relatively stable apparent erosion rates (Fig. 5). In contrast, [$^{10}$Be] in the longer Otemma UM record were found to decrease after the end of the LIA (Wetterauer et al., 2022a). Our estimates are broadly consistent with previous estimates of rockwall erosion rates in glacial landscapes in the Alps. Other studies using cosmogenic nuclides, albeit from different sources, report comparable erosion rates ranging from
0.1 to 6.4 mm yr$^{-1}$ (Wittmann et al., 2007; Sarr et al., 2019; Mair et al., 2019, 2020). Similarly, optically stimulated luminescence-derived erosion rates reach values up to 4.3 mm yr$^{-1}$ over the last ~100 years (Lehmann et al., 2020), whereas erosion rates derived from terrestrial laser scanning between the years 2005 and 2010 reach values as high as 6.5 and 8.4 mm yr$^{-1}$ (Rabatel et al., 2008; Kenner et al., 2011). Despite the similarity of our and previously estimated erosion rates, we emphasize that estimating apparent rockwall erosion rates from medial moraine [$^{10}$Be]$_{measured}$ involves uncertainties that are
not easy to quantify (Wetterauer et al., 2022a).

Apart from the assumption of isotopic steady-state that underlies our calculation (Sect. 3.1), the range of our erosion rate estimates depends on whether we account for post-depositional $^{10}$Be accumulation during downglacier debris transport (transport-corrected [$^{10}$Be]$_{rockwall}$), or not (uncorrected [$^{10}$Be]$_{measured}$). The differences are negligible for samples with high
[$^{10}$Be]$_{measured}$ but noticeable where [$^{10}$Be]$_{measured}$ are low (Table 2). Overall, our particle trajectory model involves generalized assumptions and, thus, sample ages and [$^{10}$Be]$_{transport}$ should be considered as approximations. Our temporal assignment seems to be quite robust, because even if we assume half or twice the velocity changes for years before 1977, for which we lack boulder tracking data, the modelled sample ages are mostly <5 years older, respectively younger (Table S1). Yet, we note here that the model is not capable to simulate debris transport through an icefall and that debris trajectories at Tsijiore Nouve start
at the base of the Pigne d'Arolla icefall and therefore probably underestimate the overall debris transport time. Assuming fast surface velocities across the Pigne d'Arolla icefall of up to ~70 m yr$^{-1}$ (Millan et al., 2022), sample ages at Tsijiore Nouve would be a few decades older (<40 years) than estimated, and would thus still fall into the post-LIA deglaciation period. Our estimates of supraglacial transport paths largely agree with historical photographs. At Brenay, our modelled trajectories (Fig. S3) indicate supraglacial transport below the confluence of both tributaries at all times. This is consistent with historical
photographs (Fig. 6a) and reconstructions of the glacier in 1850, which indicate that the medial moraine was already at the surface at the end of the LIA (Lambiel and Talon, 2019). At Pièce, burial and englacial transport of debris deposited before or around 1980, as suggested by our modelled trajectories (Fig. S4b), also appear to be consistent with aerial images taken in September 1983 (Fig. S5), which indicate a low ELA and re-emergence of debris further downglacier. However, exceptionally



high erosion rate estimates for Cheilon sample C5, which result from very low calculated $[^{10}\text{Be}]_{\text{rockwall}}$, suggest that the model
may overestimate $[^{10}\text{Be}]_{\text{transport}}$ at this glacier. Specifically, the simulated transport of C5 has been exclusively supraglacial (Fig. S4a), with its modelled trajectory always being slightly below the assumed ELA. We note that it is possible that the local ELA history may well differ from the assumed one, which we adopted from a regional, large-scale record (Žebre et al., 2021). In fact, samples deposited subsequently to C5 have been buried and aerial images from 1983 (Fig. S5) indicate debris deposition in the accumulation zone (at the time of image acquisition), suggesting that it is unlikely that C5 was never buried. Also, at
Tsijiore Nouve, $[^{10}\text{Be}]_{\text{transport}}$ seem to be maximum estimates, as our model does not incorporate englacial transport through the Pigne d'Arolla icefall. There, modelled debris particles experiences supraglacial transport only (Fig. S4c), even though early observations (Small and Clark, 1974) and aerial images from 1983 (Fig. S5) suggest englacial transport in the past, downglacier of the icefall base. However, in the absence of better resolved ELA, mass balance, and pre-1977 glacier velocity data, it is difficult to obtain more reliable estimates of $[^{10}\text{Be}]_{\text{transport}}$. Therefore, we consider the provided uncorrected erosion rate
estimates as minimum values, whereas our transport-corrected estimates, may instead be considered maximum values if we overestimate supraglacial transport time, as indicated by the C5 and TN samples. Nevertheless, given the mostly rather narrow ranges, except for C5 and TN samples, any trend that we observe in our apparent erosion rates is likely real and not an artefact of downglacier transport.

In addition to $^{10}\text{Be}$ accumulation during transport, the possible contribution of subglacially derived debris challenges the direct interpretation of our apparent rockwall erosion rates as actual rockwall erosion rates. Subglacially eroded debris probably has a low $[^{10}\text{Be}]$ and, if admixed with supraglacially eroded debris, would tend to reduce the concentration signal in medial moraine debris (Wetterauer et al. 2022a), especially where $[^{10}\text{Be}]$ signals of rockwall erosion are low. At Tsijiore Nouve, the two medial moraines do not directly detach from their source rockwalls, but are separated from them by the upper accumulation basin and
the Pigne d'Arolla icefall (Fig. 1g). Samples therefore possibly combine rockwall debris buried and transported englacially in the upper accumulation basin, and debris from subglacially eroded bedrock brought up by the icefall, both melting out and mixing in the ablation zone below the icefall (Small et al., 1979; Small and Gomez, 1981; Gomez and Small, 1983). Still, supraglacial sources have been suggested to be dominant, based on a higher proportion of angular medial moraine clasts (Small and Gomez, 1981). This is supported by the ridge-like topography of both moraines and their separation by an ice septum (Fig.
1g), which indicate locally enhanced supply from rockwall debris. If subglacial supply along the icefall were dominant, we would expect a more continuous debris cover over the entire ablation zone width, which is not the case. At Pièce and Cheilon, ice moving along the rockwall margin may also contribute subglacial material. Whether such material would lower the $[^{10}\text{Be}]$ of the debris by much is difficult to assess. However, rockwall debris deposition at Pièce occurs over ~200 m distance, while potential subglacial input seems to be confined to a single local point source at the easternmost rockwall margin (Fig. 6g). In
contrast, at Cheilon, the emergence of rockwalls within the glacier tributary (Fig. 6c) could allow for subglacial input along larger sections of the rockwall margin, which may explain the low concentrations measured in sample C5 (Table 1).



Since the end of the LIA, glaciers around the Pigne d'Arolla massif are retreating, exposing bedrock surfaces formerly shielded from cosmic radiation, which are now probably eroding. However, the expansion of ice-free areas is not uniform across the study sites (Table 3, Fig. 6), raising the question whether our samples with low $[^{10}Be]$ may be due to the erosion of recently uncovered bedrock with potentially low concentration (Scherler and Egholm, 2020; Wetterauer et al., 2022a). At Otemma LM, Wetterauer et al. (2022a) have found that the ice cover has shrunk significantly in patches since 1850, so that the associated patchy rockwall expansion could be the reason why $[^{10}Be]$ here show the comparatively largest spread among the deglaciation records from the massif. In contrast, between 1850 and 1973, the reconstructed rockwall expansion due to shrinking ice cover was small (<5%) at Brenay, Cheilon, Pièce, and Tsijiore Nouve, and also at Otemma UM (Fig. S1c). Since 1973, the ice-free rockwall areas have changed only slightly (~20%) at Brenay, Pièce and Tsijiore Nouve, but more significant at Cheilon and Otemma UM, where the areas have doubled (Fig. S1b). Our modelled sample ages suggest that rockwall erosion at Brenay, Cheilon and Otemma UM presumably occurred before 1973 and, thus, we expect that the associated erosion rate estimates are largely unaffected by ice cover changes. Similarly, at the younger records of Pièce and Tsijiore Nouve, post-1973 deglaciation effects should be comparatively small as rockwall areas varied only marginally. Even if our modelled age estimates were too old by a few years, it is not straightforward to provide estimates of the relative amounts of debris eroded from surfaces newly exposed since 1973 that are included in medial moraine debris and what their actual $[^{10}Be]$ is. However, we would expect continuous ice surface lowering to have notable effects, probably reflected by temporal trends, which is only observed for the long Otemma UM record. If we assume that the temporal trend at Otemma UM since 1850 reflects a pure deglaciation signal, due to the addition of debris from formerly subglacial surfaces with a $[^{10}Be]$ of zero atoms g$^{-1}$ at the same erosion rate as before, the medial moraine would require a contribution of 75% of such debris to lower the mean $[^{10}Be]_{rockwall}$ of the pre-1860 LIA value (~18×10$^3$ atoms g$^{-1}$ for O/UM6-9) to the mean $[^{10}Be]_{rockwall}$ of the post-1900 deglaciation value (~4×10$^3$ atoms g$^{-1}$ for O/UM1-3). This would correspond to a quadrupling of the debris source area and supply. Such an increase in the debris supply should also be visible in the amount of material exposed in the medial moraine. Yet, our source rockwall analysis indicate no changes in ice-free areas at Otemma UM from 1850 to 1973 (Table 3, Fig. S1c) and the downglacier narrowing of the UM between sample locations O/UM3 and O/UM8 is most probably due to the acceleration of flow velocities at the ice confluence below Petit Mont Collon (see Fig. S2c in Wetterauer et al., 2022b).

Based on the above assessments, we assume that the actual $^{10}$Be-derived rockwall erosion rates lie somewhere between our apparent minimum and maximum rockwall erosion rate estimates. The fact that within these bounds, the records are either trending or stable, rather than being randomly scattered, suggests that the operating erosion processes (see Sect. 5.2) are characterized by continuity. Source rockwall expansion into formerly glaciated terrain is small at Brenay, Cheilon, Pièce, Otemma UM and Tsijiore Nouve, but can be problematic at Otemma LM, where deglaciation is most pronounced and complex. At Cheilon and Pièce, the contribution of subglacially derived debris of low concentration is possible, probably with a greater chance of bias at Cheilon, as subglacial input could occur along larger rockwall sections. The icefall at Tsijiore Nouve likely introduces subglacial material that may lower $[^{10}Be]_{measured}$, suggesting that actual $^{10}$Be-derived rockwall erosion rates are




lower. Nevertheless, the temporal consistency of its record and the continuity of its medial moraine indicate that erosion rates remained fairly stable through time, as it is the case for most other post-LIA records, too.

### 5.2 Spatial trends in apparent rockwall erosion

The source rockwalls of the Pigne d'Arolla massif differ in their morphology (Table 3), and comparison with our apparent rockwall erosion rates, averaged over the deglaciation period, indicate some spatial trends: mean erosion rate estimates appear to be higher for smaller rockwall areas (Fig. 7a) and for steeper slopes (Fig. 7d), but vary independently of mean rockwall elevation (Fig. 7b). Differentiating by aspect, mean erosion rate estimates are higher at northwest- to northeast-facing rockwalls (Fig. 7c), but lower at southwest faces. These trends are the same for apparent minimum and maximum rockwall erosion rates.

We note that aspect and slope angle appear to be related in the area around Pigne d'Arolla: at the approximate elevation range of the studied debris source areas, ice-free north faces are the steepest, while south faces are the shallowest (Fig. 2c).

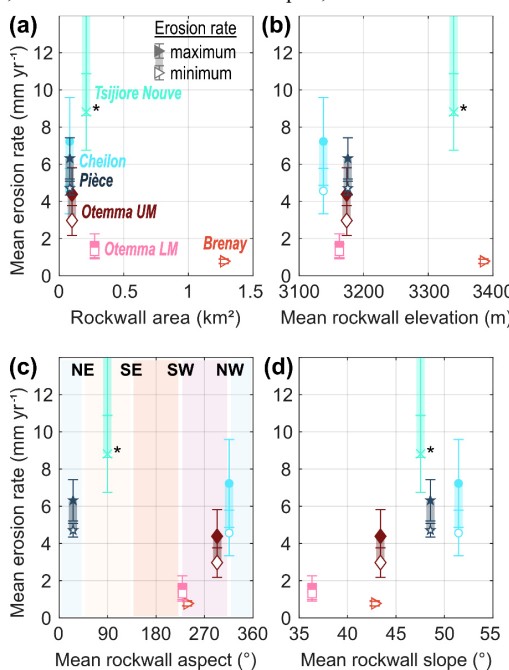

**Figure 7: Comparison of apparent mean rockwall erosion rates with respect to (a) area, (b) mean elevation, (c) mean aspect, and (d) mean slope of the 1973-exposed source rockwalls. Apparent mean erosion rates (±1σ) are depicted as shaded ranges from minimum**
**(uncorrected) to maximum (transport-corrected) estimates. For comparison, mean erosion rates here include only samples associated with the recent deglaciation period (Fig. 5), but exclude samples from the Little Ice Age (O/UM6-9). The high concentration outlier at Glacier d'Otemma (O/UM4, Fig. 5) is also excluded, but would not change the overall pattern if included. Note that mean erosion rates at Glacier de Tsijiore Nouve (*) should be treated with caution and are likely overestimated due to an unquantified subglacial bias (see Sect. 5.1), and that Glacier de Cheilon sample C5 was excluded from the maximum estimate due**
**to its unreasonable high erosion rate (see Sect. 5.1).**





At mountain ridges where asymmetry between north and south slopes has been observed, cirque erosion has been suggested to be the dominant process promoting retreat of glaciated north slopes at the expense of unglaciated south slopes during glacial periods (Oskin and Burbank, 2005; Naylor and Gabet, 2007). Our datasets indicate that asymmetry may be also observed where both north and south faces are glaciated. North of the east-west trending ridgeline between Pigne d'Arolla and Mont

Blanc de Cheilon, steeper rockwalls tower above smaller glaciers, while to the south, shallower rockwalls tower above larger glaciers (Fig. 1a,b). Besides, it appears that steeper north-facing rockwalls, although smaller in area, have higher erosion rates than the more extensive shallower south-facing rockwalls, where larger bedrock areas are exposed (Fig. 7a). In the following, we will address the relevance of (i) rockwall area, (ii) slope, and (iii) aspect as potential controls on the observed spatial trends in our apparent rockwall erosion rates.


(i) The larger a source area, the more likely it is to capture and average a variety of erosive events. At smaller areas, however, large individual events may represent a larger portion of the total area and may result in overall higher erosion rates. Nevertheless, recent rock falls in the Pigne d'Arolla massif seem small enough (Fig. 6) not to dominate an entire area or sample. According to studies that examine the size of recent rock falls in the Alps, large-volume rock falls of 100 to 1000 m$^3$

are rare, though measured over short recent time periods (e.g., Strunden et al., 2015; Hartmeyer et al., 2020), and would still only represent a fraction of our source rockwall areas (Table 3). Moreover, we could not delineate massive deposits of debris instantly released onto the ice on any of the orthoimages of the last ~40 years (swisstopo, 2022) that we used for our boulder tracking.

(ii) Erosion is typically considered a slope-dependent process (Gilbert, 1877). According to nonlinear transport laws, erosion rates linearly increase with steepening hillslopes at shallow gradients, but rapidly increase when reaching a critical hillslope angle (e.g., Roering et al., 1999, 2001). This positive correlation has been widely recognized across fluvial landscapes, where catchment-wide erosion rates derived from cosmogenic nuclides suggest a broad linear trend up to a critical angle, often around 30° (e.g., Granger et al., 1996; Binnie et al., 2007; Ouimet et al., 2009; DiBiase et al., 2012; Delunel et al., 2020). Ice-free

rockwalls in glacial landscapes, however, are much steeper and considered to be at their threshold angle (Scherler, 2014), and a similarly clear correlation may be complicated by the stochastic nature of rock falls. Still, our erosion rate estimates suggest a positive correlation with rockwall slope in the Pigne d'Arolla massif, as do erosion rates from rockwalls flanking the Mer de Glace in the Mont Blanc massif (Lehman et al., 2020).

(iii) In Alpine landscapes, spatial variability in rockwall stability and erosion is often suggested to be temperature-driven (e.g., Sass, 2005; Gruber and Haeberli, 2007; Hales and Roering, 2009; Delunel et al, 2010; Mair et al., 2020). At north faces, higher rock fall activity has been associated with higher moisture supply and deeper continued freezing, favouring damage by frost on shaded versus sunny faces (Coutard and Francou, 1989; Sass, 2005). In addition, recent rockwall destabilization has also been related to climate-induced permafrost degradation (Gruber and Haeberli, 2007), with thaw anomalies and rock fall activity





appearing to be particularly pronounced at north faces, which are accustomed to a lower inter-annual thawing depth variability
due to less direct solar radiation (Gruber et al., 2004; Sass, 2010). Besides, frost-cracking models predict rockwall erosion to
be higher where rockwalls reach into the frost-cracking window, an elevation-dependent temperature range of high frost
cracking efficiency (e.g., Walder and Hallet, 1985; Anderson, 1998; Hales and Roering, 2007), typically near the base of steep
rockwalls (Hales and Roering, 2005; Sanders et al., 2012; Scherler, 2014). Overall higher apparent rockwall erosion rates at

north faces in the Pigne d'Arolla massif, hence, could also be related to distinct aspect-related temperature conditions.
Permafrost is still most extensive at the shaded north faces of the ridgeline (Fig. S6; BAFU, 2005) and probably more
susceptible to post-LIA warming than the discontinuous permafrost occurrence to the south. Moreover, where rockwall base
elevations are similar (~3000 m at all sites except Tsijiore Nouve, Fig. 1b), temperature conditions at north faces may be more
favourable for frost cracking. It is noteworthy that, although Brenay rockwalls partially also face northwest, erosion rate

estimates are much lower than at the other north-facing sites. Here, elevation could play a role as the northwest faces at Brenay
are located high (3400-3700 m; Fig. 1b) and may experience less permafrost degradation and/or different altitudinal controls
on frost-cracking compared to the other slightly lower north-facing sites (3000-3400 m).

Based on the above assessments, we do not expect spatial trends in rockwall erosion among our study sites to be dominated
by an area effect. Rather, higher mean apparent erosion rates at steeper northwest-facing rockwalls may indicate a potential
slope and aspect control, which could be related to distinct temperature-driven destabilisation conditions. Yet, as our studied
sites are overall very steep and cover only a narrow range of slope gradients, it is not easy to resolve slope dependency in
rockwall erosion more precisely or even to generalize patterns. Moreover, it is difficult to judge whether the above suggested
aspect-related differences in permafrost and frost-cracking conditions were already as pronounced at the estimated time of

rockwall erosion. Future studies following this approach therefore may examine datasets that cover a wider range of rockwall
gradients and incorporate temporal records of land surface temperatures or frost-cracking models to further assess the role of
aspect and temperature.

## 6 Conclusions

We derived apparent rockwall erosion rates around Pigne d'Arolla in Switzerland from [$^{10}$Be] in medial moraine debris at five
adjacent valley glaciers. The total of six medial moraine records largely span 0.6 to 10.0 mm yr$^{-1}$ and cover the recent
deglaciation period (Glacier du Brenay, Glacier de Cheilon, Glacier de Pièce, Glacier de Tsijiore Nouve), back to the end of
the LIA (Glacier d'Otemma). Our analyses lead us to the following main findings:

(i) Although glacial landscapes erode stochastically, the temporal [$^{10}$Be] consistency for records from the deglaciation period
as well as the systematic [$^{10}$Be] decrease from the end of the LIA towards deglaciation, indicate that considering medial
moraines as temporal archives of rockwall erosion provides systematic results.



(ii) Post-depositional debris exposure during downglacier transport, ice cover changes across deglaciating rockwalls, and subglacially derived material introduce uncertainties that complicate the conversion of measured medial moraine [$^{10}$Be] to

rockwall erosion rates directly and require high-resolution datasets on the temporal evolution of the studied glaciers. For the majority of samples, the transport corrections derived from our boulder tracking velocities and debris trajectory modelling seem reasonable when compared to historical photographs. They are overall negligible where the measured [$^{10}$Be] are high, but more significant for lower values, yet, without affecting any observed systematic consistency or variability. Quantifying ice cover changes across the source rockwalls indicates that source area changes have been small, except for one site. Yet, the

contribution of recently deglaciated bedrock and/or subglacially derived material remains difficult to quantify, may affect some of the sites, and requires further analysis.

(iii) Temporally, [$^{10}$Be] records at Glacier du Brenay, Glacier de Cheilon, Glacier de Pièce, and Glacier de Tsijiore Nouve appear comparatively stable over the last ~100 years, as do their debris source areas for the estimated time of rockwall erosion.

This may indicate that source area changes, which are reflected in the decrease of [$^{10}$Be] in the ~200 year long record from the Glacier d'Otemma, are transient processes related to the transition from the LIA to the following deglaciation period. The absence of similar trends at the other sites may suggest that changes in source area either play a minor role, or have stabilized again after a short time period.

(iv) Source rockwalls of either northern or southern orientation across the small mountain massif of the Pigne d'Arolla are unaffected by any major climatic, tectonic or lithological differences. Higher mean apparent erosion rates at overall steeper north-facing compared to shallower southwest-facing rockwalls indicate a potential slope and aspect control on our records and may be related to rockwall erosion and destabilization affected by temperature conditions.

## 7 Data availability

The cosmogenic nuclide dataset and derived glacier surface velocities of this study will be made freely available in an accompanying data publication via GFZ Data Services if the manuscript would get accepted.

## 8 Author contribution

KW carried out sample collection and preparation, conducted the data analyses, and prepared the manuscript based on the comments and edits of DS. DS conceived and supervised the project and was the main advisor during data analyses and the

manuscript drafting.



## 9 Competing interests

The authors declare that they have no conflict of interest.

## 10 Acknowledgements

This research received funding from the European Research Council (ERC) under the European Union's Horizon 2020 research
and innovation program (grant agreement number 759639). We are grateful to Leif Anderson for support during sampling.
Stefan Heinze and Steven Binnie from the University of Cologne are thanked for performing AMS measurements. Hella
Wittmann, Cathrin Schulz and Kristina Krüger are thanked for their help and advice in sample preparation.

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
