# Peer review of "Spatial and temporal variations in rockwall erosion rates around Pigne d'Arolla, Switzerland, derived from cosmogenic 10Be in medial moraines at five adjacent valley glaciers"

_EGUsphere, 2023_

## Author Comment (AC2)

**Author response to Prof. Dr. Neil Glasser:**

Dear Professor Glasser,

thank you for taking the time to review our manuscript and for your constructive feedback. We appreciate the positive evaluation and recommendation. Our manuscript will certainly benefit from the helpful comments, and the suggested clarifications will help to improve communicating our study.

In the following we provide the review comments listed as [C1] to [C8] (black italic font) and our corresponding responses as [A1] to [A8] (blue normal font). We also quote the text additions for the revised version (inverted commas). All line numbers in this response refer to the original preprint version of the manuscript.

With kind regards,

The authors

**Comments:**

*[C1] The manuscript builds on a previous study (Wetterauer, 2022) but I was not clear how it adds to that study, other than expanding the data set from a single site (Glacier d'Otemma) to four other new sites (Glacier du Brenay, Glacier de Cheilon, Glacier de Pièce, and Glacier de Tsijiore Nouve). I would like to see a clear statement of what the addition of these new sites has added. The place to do this is probably around Line 69 of the manuscript.*

[A1] Thank you for this valid comment - we agree that it is helpful for the reader to specifically state what the additions to the previous study are. Our additions around Line 69 will be:

"This study expands the work by Wetterauer et al. (2022a) on the Glacier d'Otemma, closely resembling their approach to remain as comparable as possible. As an extension to their previous strategy of assessing the temporal evolution of rockwall erosion rates from downglacier medial moraine [$^{10}$Be] records, we now additionally integrate the comparison of records from several different glaciers with varied source rockwalls in the same area to capture possible spatial patterns in rockwall erosion. In addition to the records from Glacier d'Otemma, our new rockwall erosion dataset comprises downglacier [$^{10}$Be] records from four new study sites close by: Glacier du Brenay, Glacier de Cheilon, Glacier de Piece, and Glacier de Tsijiore Nouve. All glaciers flow down either from the northern or southern flanks of the Pigne d'Arolla but receive debris from rockwalls that differ in their exposure and morphology."

*[C2] Figure 1 could be improved. There are a lot words in the caption that should be in the legend or annotated on the various panels.*

[A2] Good point, the legend was less intuitive - thank you. We have revised Figure 1 and hope to have the legend and colours/symbols better clarified (see figure below).

*[C3] Line 188 "assuming an initial concentration of zero": can we have some discussion of the validity (or otherwise) of this statement? Does it mean you assume that the original bedrock in the rockwalls had no exposure? This seems unlikely to me as presumably these were exposed above the glaciers over the last few millenia.*

[A3] Thank you for these remarks and questions. We sense that our expression is misleading at this point. Assuming a bedrock concentration of initially 0 means that the $^{10}$Be signal within exposed rockwall surfaces only reflects the current pace of erosion. It precludes that the $^{10}$Be signal is a mixture from nuclides due to erosion and nuclides inherited due to repeated ice coverage in-between different episodes of bedrock exposure with possibly different $^{10}$Be signals. However, such a simple exposure history may not apply in glacial landscapes, where the exposure of rock alternates with ice coverage. This is part of the reason why we refer to our rate estimates as "apparent" (Lines 192ff). We try to asses and discuss this aspect further by quantifying and evaluating the impact of areal ice cover changes on our rate estimates (Section 5.1). Nevertheless, quantifying erosion rates from cosmogenic nuclide concentrations with Equation 1 requires this assumption.

To avoid misunderstandings, we will omit the formulation "assuming an initial concentration of zero" from Line 188 in the revised manuscript, especially since this assumption is implied in the cosmogenic steady state assumption (Line 185) anyway.

*[C4] Line 208: I would like to see a statement about how representative (or otherwise) the samples are. How were the samples collected and how did you decide where to collect them? How did you ensure they represent the local environment at each point of the moraines?*

[A4] Thank you for this comment. In the revised version, Lines 208 and 209 will be replaced by the following more detailed descriptions:

"As rockwalls erode by stochastic processes and individual bedrock samples may have different [$^{10}$Be], rockwall erosion rate records should reflect outcrop-scale average rates and thus amalgamate [$^{10}$Be] across the entire source rockwall area, to avoid episodic effects on single measurements (Small et al., 1997). Therefore, each sample is a 3-4 kg weighing amalgamation of supraglacial clasts from coarse sand to pebble size (~1-30 mm), randomly hand-scraped across medial moraine surface areas that cover 5-30 m along a moraine by the entire moraine width. In addition, to minimize subjective selection of sample locations, samples from the same medial moraine were taken at regular downglacier intervals."

*[C5] Line 257 onwards: Please add an uncertainty estimate for the velocities here and elsewhere.*

[A5] This is indeed a valid suggestion and we agree that the uncertainty on our velocity reconstructions needs to be addressed - we admit, however, that providing precise ± uncertainty estimates per tracked boulder is not trivial. Uncertainties on the boulder velocities basically stem from three main sources: the positional accuracy of the orthoimages used for boulder tracking that varies throughout time (see Lines 263ff), the subjective tracking of a medial moraine boulder across orthoimages, and the temporal interpolation between full years due to the unavailability of exact image acquisition dates. In addition, there exist uncertainties on velocity estimates prior to 1977, where we lack boulder tracking data. We assessed the impact of velocity uncertainties by simulating downglacier particle trajectories based on multiplying our estimated velocities by both 0.5 and 2.

Even such large changes in velocities however, have a limited effect on our travel time estimates (see also our answer [A6] to the following comment [C6]).

Nevertheless, for the revised version we include a descriptive paragraph to our supplementary material where we try to provide an average uncertainty estimate on the derived boulder velocities using Gaussian error propagation.

**[C6]** *Line 305 "We acknowledge that our reconstruction of glacier surface velocities and elevations contains several sources of unquantified uncertainties": can you say more about how these unquantified uncertainties are a limitation and how this then plays out? Where is it later discussed? (mentioned again on Line 405).*

**[A6]** Unfortunately, we are not able to provide more precise ± velocity uncertainty estimates than outlined in our response [A5] to comment [C5] above. We acknowledge that these are limitations in our study and try to discuss how these may play out in the first discussion chapter (Section 5.1) on the "appearance" of the erosion rate estimates (Lines 410-413). By modelling downglacier transport with half/twice our flow velocity estimates (Table S1) we try to assess how different velocity estimates prior to 1977 would affect debris particle transport time/ sample ages. In most cases, differences on sample ages are small, <5 years. Nevertheless, we provide our rate estimates twofold, transport-corrected and uncorrected, to show that accounting for downglacier transport time or not does not change but only shift the overall temporal patterns in erosion rates along our downglacier records or the overall spatial patterns among different sites.

**[C7]** *Line 576: What does "provides systematic results" mean?*

**[A7]** We agree, this phrase is too general. We hope to clarify this first conclusion statement from Lines 574-576 in the revised version by:

"Although glacial landscapes typically erode stochastically, which could make random variability along temporal records likely, our medial moraine [10Be] records are temporally consistent for the deglaciation period and decrease systematically from the end of the LIA towards deglaciation. Whereas large-volume rock falls are typically identifiable by the additional volume of rock delivered to the glacier, the morphology of the studied medial moraines varies only gradually and is more likely to reflect gradual variations in ice dynamics or rockwall erosion. This suggests that the medial moraines from the Pigne d'Arolla massif indeed act as archives of rockwall erosion."

**[C8]** *Conclusion iv (line 596): I am not sure you can say this (that the source walls are unaffected by any major climatic, tectonic or lithological differences) because the temporal records you have derived are so short by comparison.*

**[A8]** We apologise, this wording was expressed in a misleading way - thank you for pointing this out. We intended to say that the observed differences in [10Be] (respectively erosion rates) among our medial moraine records from the deglaciation period are likely not due to major lithological/climatic/tectonic differences among the studied sites, which could arise if the sites would not be part of the small and localized Pigne d'Arolla massif only, but from various settings spread

across the Alps. We hope to have this improved by rephrasing Lines 595-598 in the revised version to:

"The choice of the small and localized massif around Pigne d'Arolla, with its similar rockwall lithology, facilitates the relative spatial comparison of its debris source areas without major bias arising from comparing sites with different tectonic or climatic settings or pronounced lithological differences. Around Pigne d'Arolla, higher mean apparent erosion rates at overall steeper north-facing compared to shallower southwest-facing rockwalls indicate a potential slope and aspect control on our records and may be related to rockwall erosion and destabilization affected by temperature conditions."

**References:**

Small, E.E., Anderson, R.S., Repka, J.L., and Finkel, R.: Erosion rates of alpine bedrock summit surfaces deduced from in situ $^{10}$Be and $^{26}$Al, Earth Planet. Sc. Lett., 150, 413-425, https://doi.org/10.1016/S0012-821X(97)00092-7, 1997.

Swisstopo: Federal Office of Topography swisstopo, https://map.geo.admin.ch, last access: 25 May 2023.

Wetterauer, K., Scherler, D., Anderson, L.S., and Wittmann, H.: Temporal evolution of rockwall erosion rates derived from cosmogenic nuclide concentrations in the medial moraines of Glacier d'Otemma, Switzerland, Earth Surf. Proc. Land., 47, 2437-2454, https://doi.org/10.1002/esp.5386, 2022

**Revised legends in Figure 1:**

[Figure]

**Figure 1:** Pigne d'Arolla massif, Switzerland. (a) Orthoimage showing the five adjacent glacier catchments and their downglacier medial moraine sample locations. The respective associated source rockwalls are outlined (orthoimage from 2017 by swisstopo, 2022). (b) Hillshade image of the same area as in (a) with glacier extents in 2016 and 1973 after Linsbauer et al. (2021) and Müller et al. (1976), respectively. The snowline of the 2017 orthoimage is indicated as approximation of the recent equilibrium line altitude. Hillshade and 100 m spaced contour lines are based on the DEM SRTM GL1 (NASA SRTM, 2013). White rectangles indicate map extents shown in Fig. 3. Note the east-west trending ridgeline between Pigne d'Arolla and Mont Blanc de Cheilon following closely the ~3500 m contour line. (c-g) Field photographs showing medial moraines, approximated sample locations and associated source rockwalls of (c) Glacier du Brenay, (d) Glacier de Cheilon, (e) Glacier d'Otemma (samples from Wetterauer et al., 2022a), (f) Glacier de Pièce, and (g) Glacier de Tsijiore Nouve.